# Bioactive Platinum(IV) Complexes Incorporating Halogenated Phenylacetates

**DOI:** 10.3390/molecules27207120

**Published:** 2022-10-21

**Authors:** Angelico D. Aputen, Maria George Elias, Jayne Gilbert, Jennette A. Sakoff, Christopher P. Gordon, Kieran F. Scott, Janice R. Aldrich-Wright

**Affiliations:** 1School of Science, Western Sydney University, Locked Bag 1797, Sydney, NSW 2751, Australia; 2Ingham Institute, Sydney, NSW 2170, Australia; 3Calvary Mater Newcastle Hospital, Newcastle, NSW 2298, Australia

**Keywords:** chemotherapy, cisplatin, PHEN*SS*, 56ME*SS*, platinum(II), platinum(IV), phenylacetate, lipophilicity, cytotoxicity, ROS

## Abstract

A new series of cytotoxic platinum(IV) complexes (**1**–**8**) incorporating halogenated phenylacetic acid derivatives (4-chlorophenylacetic acid, 4-fluorophenylacetic acid, 4-bromophenylacetic acid and 4-iodophenylacetic acid) were synthesised and characterised using spectroscopic and spectrometric techniques. Complexes **1**–**8** were assessed on a panel of cell lines including HT29 colon, U87 glioblastoma, MCF-7 breast, A2780 ovarian, H460 lung, A431 skin, Du145 prostate, BE2-C neuroblastoma, SJ-G2 glioblastoma, MIA pancreas, the ADDP-resistant ovarian variant, and the non-tumour-derived MCF10A breast line. The in vitro cytotoxicity results confirmed the superior biological activity of the studied complexes, especially those containing 4-fluorophenylacetic acid and 4-bromophenylacetic acid ligands, namely **4** and **6**, eliciting an average GI_50_ value of 20 nM over the range of cell lines tested. In the Du145 prostate cell line, **4** exhibited the highest degree of potency amongst the derivatives, displaying a GI_50_ value of 0.7 nM, which makes it 1700-fold more potent than cisplatin (1200 nM) and nearly 7-fold more potent than our lead complex, **56ME*SS*** (4.6 nM) in this cell line. Notably, in the ADDP-resistant ovarian variant cell line, **4** (6 nM) was found to be almost 4700-fold more potent than cisplatin. Reduction reaction experiments were also undertaken, along with studies aimed at determining the complexes’ solubility, stability, lipophilicity, and reactive oxygen species production.

## 1. Introduction

C*is*-diamminedichloroplatinum(II) (cisplatin) (Figure 1) is the most celebrated metal-based chemotherapeutic in the history of modern medicine. The discovery of its anticancer effects was not only opportune [1,2,3], especially during a period when cancer prevention was desperately needed, but also because medical oncology did not yet exist as a clinical specialty [4]. Despite the widespread scepticism on the application of anticancer drugs (or poison, as they were called), cisplatin became the first platinum(II) coordination complex to be approved for the systemic treatment of cancer [5]. The clinical success of cisplatin shed light on the structure–activity relationship (SAR) dogma that was popularised by Cleare and Hoeschele [6], a drug design strategy devised to develop additional chemotherapeutics utilising cisplatin as a scaffold. As a result, *trans*-*L*-(1*R*,2*R*-diaminocyclohexane) oxalatoplatinum(II) (oxaliplatin) and *cis*-diammine(1,1-cyclobutanedicarboxylato) platinum(II) (carboplatin) (Figure 1) were developed and later approved as additional platinum-based chemotherapeutics [6,7]. Although proven highly efficient against multiple cancer types [8,9,10], platinum(II) drugs have undesirable attributes that significantly impact the quality of life for cancer patients [11,12,13,14].

Deviating from the SAR design that created cisplatin derivatives, our research group has previously reported a selection of structurally distinct platinum(II) complexes of the type **[Pt^II^(H_L_)(A_L_)]^2+^**, where H_L_ is the heterocyclic ligand (i.e., 1,10-phenanthroline (phen) or 5,6-dimethyl-1,10-phenanthroline (5,6-Me_2_phen)) and A_L_ is a chiral ancillary ligand (1*S*,2*S*-diaminocyclohexane (*SS*-DACH or DACH)) (Figure 2) [15,16,17,18,19,20]. Amongst all complexes investigated within the class, [Pt^II^(5,6-Me_2_phen)(*SS*-DACH)]^2+^ (**56ME*SS***) (Figure 2) proved to be the most biologically active, demonstrating superior anticancer killing properties in multiple human cancer cell lines compared to platinum(II) drugs [15,16,17,18,20,21]. Its analogue, [Pt^II^(phen)(*SS*-DACH)]^2+^ (**PHEN*SS***) (Figure 2), also showed significant potency in other human cancer cell lines compared to cisplatin and its derivatives [16,17,19,20,21]. Interestingly, the mechanism of action of our lead complex, **56ME*SS***, differs from the covalent binding mechanism of clinically used platinum(II) drugs [22]. **56ME*SS*** interferes with the mitochondria, cytoskeletal proteins, and to a lesser extent nuclear DNA [23].

Despite evidence that these unconventional platinum(II) complexes exhibit potent cytotoxicity in vitro, there has been mixed evidence that this activity is translated into in vivo studies. For example, BD-IX rats with peritoneal carcinomatosis were treated with **56ME*SS*** via intravenous and intraperitoneal methods; this treatment did not elicit a tumour suppression response and at pharmacological doses induced nephrotoxicity [24]. By contrast, in a separate study, mice bearing PC3 (prostate carcinoma) tumour xenografts treated with **PHEN*SS*** or cisplatin both demonstrated a comparable decrease in mean tumour weight in relation to the control group [19]. Importantly, with no obvious signs of toxicity observed in mice treated with **PHEN*SS***, half of the cisplatin-treated mice perished by day 20. Because of this discrepancy between in vitro and in vivo antitumor effects of these unconventional platinum(II) complexes, their corresponding platinum(IV) complexes have been explored as a strategy to harness their potential [21,25,26,27,28,29,30]. Preliminary in vivo data on murine LLC models, administered orally, demonstrated that unconventional platinum(IV) complexes outperformed cisplatin [30].

What makes platinum(IV) complexes attractive as a drug design strategy is their octahedral geometry that allows the inclusion of two additional ligands through coordination of the axial positions [31]. The additional coordination positions offer the ability to design combinatorial therapies in a single complex, which is potentially more effective than the precursor platinum(II) complex. Specifically, platinum(IV) complexes are proposed to be stable in the bloodstream due to their low spin d^6^ electronic configuration [32]. It is only once platinum(IV) complexes cross into cancer cells that the platinum(IV) undergoes reduction to the biologically active species with the release of the axial ligands [33], and this makes them effective prodrugs. Normally, reduction occurs in the presence of biological reductants such as ascorbic acid (AsA) or glutathione (GSH) [33,34]. This unique advantage allows platinum(IV) complexes to potentially avoid off-target biomolecules in the body, resulting in fewer side effects and increasing drug selectivity [35]. The effectiveness of this characteristic has been explored with *bis*-(acetate)-ammine dichloro-(cyclohexylamine) platinum(IV) (satraplatin), *cis*,*trans*,*cis*-dichlorodihydroxo-*bis*-(isopropylamine) platinum(IV) (iproplatin), and tetrachloro(1*R*,2*R*-diaminocyclohexane) platinum(IV) (tetraplatin) (Figure 3). Despite the potential of this approach, no platinum(IV) complexes of this type have been approved for clinical use due to either lack of activity or failure to demonstrate an overall improvement in survival [36].

Recent studies have focused on developing platinum(IV) complexes in combination with a diverse class of bioactive inhibitors that perturb cancer development, and interestingly, cisplatin remains a popular scaffold [37,38,39,40,41,42]. Histone deacetylase inhibitors, such as 4-phenylbutyric acid and valproic acid (Figure 4), are a class of anticancer agents, which have been coordinated with cisplatin because of their ability to expose DNA to platination [43]. Likewise, non-steroidal anti-inflammatory drugs, such as indomethacin, aspirin, ibuprofen, and naproxen (Figure 4), have also been conjugated with cisplatin to produce prodrugs that primarily target tumour-associated inflammation [44]. Additionally, dichloroacetic acid (DCA) (Figure 4) is a pyruvate dehydrogenase kinase inhibitor that has been extensively reported in the literature because of its effectiveness in suppressing the Warburg effect (a cancer hallmark) and its synergism with cisplatin [45,46,47,48,49,50]. The success of DCA as co-treatment with other chemotherapeutics has also been widely reported in the literature, despite it not being FDA-approved [46,50,51,52,53,54]. Furthermore, the DNA alkylating agent, chlorambucil (Figure 4), has also been coordinated with cisplatin and oxaliplatin to generate dual-DNA targeting prodrugs [55,56,57,58].

With the many bioactive axial ligands that have been studied with platinum, a naturally occurring auxin substance called phenylacetic acid (PAA) caught our interest (Figure 5). Apart from being a fundamental block of multiple pharmaceutical intermediates, PAA is also a promising non-toxic drug for cancer treatment [59,60,61,62]. It was suggested that by preserving the aromatic ring and carboxylic group of PAA while adding other structural functionality, more potent derivatives might be generated [59]. In light of this, we introduced a series of halogenated PAA derivatives, including 4-chlorophenylacetic acid (4-CPA), 4-fluorophenylacetic acid (4-FPA), 4-bromophenylacetic acid (4-BPA), and 4-iodophenylacetic acid (4-IPA) (Figure 5), as axial ligands to the cores of our reactive platinum(IV) scaffold, **[Pt^IV^(H_L_)(A_L_)(OH)_2_]^2+^**. This design approach was intended to create multimodal prodrugs and to investigate if the halogens on the axially coordinated phenylacetic acids (chlorine (Cl), fluorine (F), bromine (Br) and iodine (I)) would impact the solubility, stability, reduction property, lipophilicity, overall cytotoxicity, and reactive oxygen species (ROS) potential of the prodrugs.

Amongst the selected axial ligands, 4-CPA was of primary interest because previous studies highlighted its effectiveness against estrogen-induced mammary tumorigenesis [63,64,65]. Estrogen and its receptor are critical factors in both normal breast development and the progress of breast carcinomas [66]. Further evidence suggests that estrogen production in breast tissues is mainly due to the overexpression of aromatase enzymes, especially in the case of postmenopausal breast cancer [67,68]. Aromatase is crucial in estrogen biosynthesis, as it catalyses the aromatization of androgen to form estrogen. Increased aromatase enzyme activity is also found in tumour-expressing breast cancer cells, suggesting such enzymes are viable therapeutic targets [66,68]. Treating estrogen-sensitive breast cancer with current therapies is challenging, as resistance develops after prolonged administration and can involve complications [69]. From this, Sidell et al. studied the efficacy of 4-CPA as a potent aromatase inhibitor and demonstrated a novel mechanism of action by which it antagonizes estrogen signalling [63]. Interestingly, this differs from the mechanism of action of classical anti-breast cancer drugs such as tamoxifen. A prototype platinum(IV) prodrug bearing 4-CPA is expected to express the attributes of both moieties and provide evidence for the effectiveness of this strategy for treating estrogen-sensitive breast cancer.

Here we report the synthesis, characterisation, and biological investigations of mono-substituted platinum(IV) complexes classified as **[Pt^IV^(H_L_)(A_L_)(X)(OH)]^2+^**, where **X** represents the halogenated PAA axial ligands, 4-CPA, 4-FPA, 4-BPA, and 4-IPA: [Pt^IV^(phen)(*SS*-DACH)(4-CPA)(OH)]^2+^ (**[PHEN*SS*(IV)(4-CPA)(OH)]^2+^** = **1**), [Pt^IV^(5,6-Me_2_phen)(*SS*-DACH)(4-CPA)(OH)]^2+^ (**[56ME*SS*(IV)(4-CPA)(OH)]^2+^** = **2**), [Pt^IV^(phen)(*SS*-DACH)(4-FPA)(OH)]^2+^ (**[PHEN*SS*(IV)(4-FPA)(OH)]^2+^** = **3**), [Pt^IV^(5,6-Me_2_phen)(*SS*-DACH)(4-FPA)(OH)]^2+^ (**[56ME*SS*(IV)(4-FPA)(OH)]^2+^** = **4**), [Pt^IV^(phen)(*SS*-DACH)(4-BPA)(OH)]^2+^ (**[PHEN*SS*(IV)(4-BPA)(OH)]^2+^** = **5**), [Pt^IV^(5,6-Me_2_phen)(*SS*-DACH)(4-BPA)(OH)]^2+^ (**[56ME*SS*(IV)(4-BPA)(OH)]^2+^** = **6**), [Pt^IV^(phen)(*SS*-DACH)(4-IPA)(OH)]^2+^ (**[PHEN*SS*(IV)(4-IPA)(OH)]^2+^** = **7**), and [Pt^IV^(5,6-Me_2_phen)(*SS*-DACH)(4-IPA)(OH)]^2+^ (**[56ME*SS*(IV)(4-IPA)(OH)]^2+^** = **8**) (Figure 6). The structure and purity of the complexes were confirmed through high-performance liquid chromatography (HPLC), nuclear magnetic resonance (^1^H-NMR; ^19^F-NMR; two-dimensional correlation spectroscopy (2D-COSY); heteronuclear multiple-quantum correlation (^1^H−^195^Pt-HMQC)), ultraviolet-visible (UV), circular dichroism (CD), and high-resolution electrospray ionization mass spectrometry (ESI-MS). Complexes **1**–**8** were tested against eleven human cancer cell lines representative of HT29 colon, U87 glioblastoma, MCF-7 breast, A2780 ovarian, H460 lung, A431 skin, Du145 prostate, BE2-C neuroblastoma, SJ-G2 glioblastoma, MIA pancreas and the ADDP-resistant ovarian variant, and the non-tumour derived MCF10A breast line. In addition to solubility, stability, lipophilicity, and ROS measurements, the reduction properties of **1**–**8** were investigated by ^1^H-NMR and one-dimensional ^195^Pt-NMR (1D-^195^Pt-NMR).

## 2. Results and Discussion

### 2.1. Synthesis and Characterisaion

All NHS esters were synthesised as previously described with minor modifications [70,71]. Minor modifications refer to solvent choice, reaction time, and modifying extraction or isolation steps to obtain the products with higher yields. Each starting acid was reacted with a dehydrating agent, DCC in C_3_H_6_O or DCM to afford a cloudy solution that is primarily due to the presence of an unwanted by-product, dicyclohexylurea (DCU) (Figure 7). The solutions were filtered either by gravity filtration or syringe filtration, collected, and reduced to dryness to afford a crystalline product. Syringe filtration was more efficient than gravity filtration. In some instances, the resulting products were isolated as oils. The products were used without purification in the following reactions. An average yield of ~80% was recorded for all of the NHS esters. Characterisation to confirm the successful synthesis of the esters included HPLC (Appendix A), NMR (Appendix A), and ESI-MS (Appendix A) experiments.

The precursor platinum(II) (**PHEN*SS*** and **56ME*SS***) and platinum(IV) (**PHEN*SS*(IV)(OH)_2_** and **56ME*SS*(IV)(OH)_2_**) complexes were synthesised as previously described [21,71]. The synthesis of the desired products, **1**–**8**, was initially undertaken using the methods reported in the literature [30,72,73]. Unfortunately, these methods did not produce the intended complexes reliably (i.e., poor yield, unwanted by-products, and product isolation difficulty); therefore, method development was required to obtain the most efficient method for isolating the complexes [71]. To prepare **1**–**8**, the appropriate precursor, **[Pt^IV^(H_L_)(A_L_)(OH)_2_]^2+^**, was reacted with 2–3.5 mol eq. of the NHS esters in DMSO and left stirring for 72 h at room temperature in the dark (Figure 8). Each reaction was washed with excess Et_2_O to afford two layers: a colourless and an oily brown layer. The colourless layer was discarded while the oily brown layer was collected and diluted with a minimum amount of MeOH, followed by the addition of excess Et_2_O to afford a beige precipitate. The precipitate was collected and washed with excess C_3_H_6_O to remove any unreacted or excess NHS esters and acids. Further purification using flash chromatography was required to obtain higher purity. Characterisation to confirm the successful synthesis of **1**–**8** included HPLC (Appendix A), NMR (**1**: Appendix A; **2**: Appendix A; **3**: Appendix A; **4**: Appendix A; **5**: Appendix A; **6**: Appendix A; **7**: Appendix A; **8**: Appendix A), UV (Appendix A), CD (Appendix A), and ESI-MS (Appendix A) experiments. All experimental yields, HPLC peak areas (%), retention times (T_R_), and mass-to-charge ratios (*m*/*z*) of the studied complexes are summarised in Table 1.

#### 2.1.1. H-NMR, 2D-COSY, and ^1^H−^195^Pt-HMQC Spectral Assignment

A summary of the ^1^H-NMR, ^19^F-NMR, and ^1^H−^195^Pt-HMQC data of **1**–**8** is presented in Table 2. A total of 10–11 peaks, including the solvent peak, were recorded for the studied complexes. No amine protons were observed due to proton exchange with D_2_O. Elucidating **1** as an example, the ^1^H-NMR spectrum displays slightly shifted downfield multiplicity originating from the phen and DACH protons, which is attributed to the coordination of 4-CPA to the platinum (Figure 9). The doublets at 9.22 and 9.17 ppm, with calculated *J*-coupling constants of 5.6 and 5.5 Hz, were assigned to aromatic protons H2 and H9, respectively. H4 and H7 resonated as a triplet at 9.03 ppm, while H3 and H8 protons appeared as a multiplet as shown in Figure 9. These resonances are typically observed as doublets for the platinum(II) and platinum(IV) precursors, **PHEN*SS*** and **PHEN*SS*(IV)(OH)_2_**, respectively [21]. We postulate that the differences in multiplicity are a consequence of axial coordination of the 4-CPA ligand, thereby shifting the chemical resonances more downfield due to a deshielding effect, and this was also evident in the ^1^H-NMR spectra of **2** (Appendix A).

Two sharp doublets were also observed at 6.53 and 6.28 ppm, with the same calculated *J*-coupling constant of 8.3 Hz (Figure 9). These sharp doublets were from the aromatic protons of 4-CPA, represented by a, b, c, and d. This indicates that the 4-CPA ligand was successfully coordinated to the intended scaffold. During the preliminary stages of assigning peaks, it was not clear where the peak for the methylene protons (e) of 4-CPA resonated. Although, by referring to the multiplet at 3.13 ppm in Figure 9 where DACH protons, H1′ and H2′, resonated, a proton integration of 4 was recorded indicating that two extra protons were present. As a result, it was concluded that e overlapped with the peaks of H1′ and H2′. To further confirm this, 2D-COSY was undertaken (Appendix A). From the expanded 2D-COSY spectra of **1** (Figure 10), there is long-distance coupling between e and H1′ and H2′ at 3.13 ppm. This was also observed in the 2D-COSY of **2** (Appendix A).

^1^H−^195^Pt-HMQC experiments were also acquired to further confirm that the 4-CPA ligand has been successfully coordinated and only occupied one axial position. Typically, platinum(II) resonates at −2800 ppm while platinum(IV) resonates at 400 ppm, as previously reported [21]. In the ^1^H−^195^Pt-HMQC spectrum of **1** (Figure 11), three peaks were observed at 523 ppm, highlighting the correlation of the protons originating from the heterocyclic ligand (phen), as well as the 4-CPA and DACH ligands. The cross-coupling of the aromatic protons, H2 and H9 (9.22 and 9.17 ppm), and H3 and H8 (8.20 ppm) with the ^195^Pt peaks at 523 ppm indicate correlation. The cross-coupling between 3.13 and 523 ppm observed in the ^1^H−^195^Pt-HMQC of **1** was deduced to be e protons of 4-CPA coupling with the platinum. Furthermore, the same was observed in the ^1^H−^195^Pt-HMQC of **2**; the only notable discrepancy is the ^195^Pt resonance occurring at 531 ppm (Appendix A).

The ^1^H-NMR, 2D-COSY and ^1^H−^195^Pt-HMQC spectra of the remaining complexes, **3**–**8** (**3**: Appendix A; **4**: Appendix A; **5**: Appendix A; **6**: Appendix A; **7**: Appendix A; **8**: Appendix A), also exhibited similar resonance patterns to those demonstrated by **1** and **2**. Notably, for **3** and **4**, which contain the 4-FPA ligand, the signals originating from the aromatic protons of 4-FPA (a, b, c, and d) overlapped, which is proposed to be induced through the electronegativity of the F atom (**3**: Appendix A; **4**: Appendix A). ^19^F-NMR was also carried out as an additional study for **3** and **4**, to confirm further the presence of the F atom in the complexes (**3**: Appendix A; **4**: Appendix A). Furthermore, the ^1^H-NMR spectrum of the studied complexes (**1**–**8**) was stacked together, as presented in Figure 12, for comparison purposes. It is evident that the signals originating from the heterocyclic ligands (phen and 5,6-Me_2_phen) and DACH are consistent amongst the complexes. Of further note is the distance of signal splitting in the aromatic region (6–7 ppm), wherein the coordinated ligands resonated (Figure 12). Evidently, the greater the size of the halogen that is on the PAA ligand and the less electronegative it is, the greater the splitting is observed, as highlighted in Figure 12.

#### 2.1.2. UV and CD Measurements

The electronic transitions detected in the UV measurements are consistent with published examples [21,25,26,27,71,74]. To attain consistency and accuracy, the UV spectra of complexes **1**–**8** (Appendix A) were obtained by titrations for each complex in triplicate. Molar extinction coefficients were also calculated based on a generated plot curve of absorbance against concentration using the Beer–Lambert law equation. Standard deviations were also calculated. In addition to UV absorption, CD measurements were also undertaken to confirm the retention of chirality of the ancillary ligand, *SS*-DACH. The obtained CD spectra (Appendix A) are also consistent with published examples [21,25,26,27,71,74]. All characteristic peaks in the UV and CD spectra of the complexes are summarised in Table 3.

The influence of the halogens in the axial ligands of **1**–**8** produced slight differences or patterns based on their UV spectra (Appendix A). The results obtained from the UV measurements demonstrated both π–π* transitions and metal-to-ligand charge transfer interactions. The studied complexes are aromatic-rich because of the presence of heterocyclic ligands (phen and 5,6-Me_2_phen), which exhibit ligand-centred π–π* transitions. The UV spectra of platinum(IV) derivatives that contain phen, such as **1**, **3**, **5** and **7**, exhibited similar absorption bands (Figure 13). However, **3** demonstrated a strong absorption band at 204 nm, which was not observed for **1**, **5**, and **7**, as shown in Figure 13. This band was also observed in their platinum(IV) precursor complex, **PHEN*SS*(IV)(OH)_2_** [27]. Another observation was the slight bathochromic effect, or red shift exhibited at 245 nm for **7**. These changes suggest that the phen derivatives containing F and I atoms had influenced slight observable shifts (**3**: strong band at 204 nm; **7**: bathochromic shift at 245 nm) compared to the phen derivatives containing Cl and Br (**1** and **5**, respectively). Furthermore, the UV spectra of platinum(IV) derivatives that contain 5,6-Me_2_phen (**2**, **4**, **6** and **8**) also exhibited similar absorption bands, specifically the weak shoulder-like features between 220 and 250 nm followed by prominent bands between 280 and 290 nm (Figure 14). A prominent peak was observed for **2** at 287 nm, which is broader compared to what **4**, **6**, and **8** exhibited at that wavelength (Figure 14). Interestingly, the slight shifts between 220 and 250 nm shown in Figure 14 may be attributed to the halogens; however, these shifts were not demonstrated for the phen derivatives, **1**, **3**, **5**, and **7**. Of further note, at lower wavelengths (circa 207 nm), **2**, **4**, **6**, and **8** did not produce the bands typically observed for their platinum(IV) precursor complex, **56ME*SS*(IV)(OH)_2_** [27]. Again, this confirms that the halogens slightly influenced the absorption bands recorded for the studied complexes.

Chirality is crucial to the biological activity of the studied complexes. It has been previously confirmed that by replacing the chiral ancillary ligand *SS*-DACH with its enantiomer *RR*-DACH in the structure of our complexes, it significantly impacts on the cytotoxicity [17,18,20,75]. For example, in the murine leukaemia cell line (L1210), **56ME*RR*** and **56ME*SS*** elicited GI_50_ values of 460 and 9 nM, respectively, demonstrating a 51-fold difference in cytotoxicity [75]. In the same cell line, an almost 12-fold difference in cytotoxicity was demonstrated between **PHEN*RR*** (1500 nM) and **PHEN*SS*** (130 nM) [17].

CD measurements were undertaken in this study mainly to confirm that the *SS*-DACH of the studied complexes has been retained. Complexes **1**–**8** produced strong negative absorptions bands at lower wavelengths and weak positive bands at higher wavelengths (Appendix A). These results are consistent with published examples of platinum(IV) complexes, as they also exhibit negative absorption bands at lower wavelengths and positive absorption bands at higher wavelengths [21,25,26,27,71,74]. By contrast, platinum(IV) complexes containing *RR*-DACH generally demonstrate positive absorption bands at lower wavelengths and negative absorption bands at higher wavelengths [27,74].

### 2.2. Solubility and Stability

In drug development, solubility and stability are important parameters that have a critical role in drug safety and effectiveness [76,77,78]. Since the complexes presented in this study, **1**–**8**, incorporate axial ligands that contain different halogens (F, Cl, Br, and I), we investigated their solubility and stability in aqueous solution to see the impact of the halogens. Complexes **1**–**8** were found to be soluble in d.i.H_2_O at room temperature (Table 4). Interestingly, slight variations in their solubility were demonstrated, as shown in Table 4. Apart from **1**, the derivatives containing F and Cl (**2**, **3**, and **4**) were more soluble than those derivatives containing Br and I (**5**, **6**, **7**, and **8**) in d.i.H_2_O. Furthermore, the 5,6-Me_2_phen derivatives (**2**, **4**, **6**, and **8**) were found to be more soluble than the phen derivatives (**1**, **3**, **5**, and **7**) in d.i.H_2_O, which may also be directly influenced by the methylation of the heterocyclic ligand. While the methylation of the heterocyclic ligand and the halogenated axial ligands did alter the solubility of the complexes, their effects were not significant.

The stabilities of **1**–**8** were also evaluated using HPLC. The complexes were dissolved in 10 mM of PBS (~7.4 pH) and incubated at 37 °C to mimic physiological conditions. The process was followed for 36 h. Based on the HPLC chromatograms obtained (Appendix A), no significant reduction to their platinum(II) and platinum(IV) precursor complexes was observed, confirming their stability in solution for 36 h. Overall, the halogenated axial ligands and the methylation of the heterocyclic ligand did not impact the stability of the complexes.

### 2.3. Lipophilicity

For lipophilicity measurements, HPLC was utilised instead of the traditional shake-flask method, as described previously [21,26,29]. Recent studies have also reported the efficiency of HPLC in measuring the lipophilicity of compounds [79,80]. Each complex was eluted under several isocratic ratios. A standard curve was generated to calculate the log value of the capacity factor (log k) according to Equation (1). Following Equation (2), the percentage of organic solvent (CH_3_CN) in the mobile phase from each isocratic run was plotted against log k_w_ (Appendix A). Log k_w_ is the chromatographic lipophilicity index or measure of lipophilicity of the complex, so a higher value corresponds to greater lipophilicity [26,81,82]. Because the selected axial ligands in this study contain F, Cl, Br, and I, it was only appropriate to determine the influence of the halogens on the overall lipophilicity of the platinum(IV) complexes. According to the calculated log k_w_ values (Table 5), the phen derivatives (**1**, **3**, **5**, and **7**) demonstrated lower lipophilicity values compared to their 5,6-Me_2_phen counterparts (**2**, **4**, **6**, and **8**) as expected, and this trend is consistent with the literature data [21,26,29]. This only indicates that the presence and absence of methyl groups in the heterocyclic ring system of the complexes play a role in altering lipophilicity. The most lipophilic complex was the 5,6-Me_2_phen derivative, **8**, which contains 4-IPA, demonstrating a log k_w_ of 1.19. Conversely, the least lipophilic complex was the phen derivative containing 4-FPA, **3**, with a log k_w_ of 0.25. Furthermore, for the phen derivatives, the order of increasing lipophilicity was **3** > **1** > **5** > **7**, which corresponds to the order of increasing size of the halogens, F > Cl > Br > I. This pattern was also observed with the 5,6-Me_2_phen derivatives (**4** > **2** > **6** > **8**), thereby confirming that the halogens increased the order of lipophilicity accordingly based on their size.

### 2.4. Reduction Experiments

Platinum(IV) complexes are reduced to their platinum(II) precursors by intracellular biological reducing agents such as AsA or GSH in tumour cells [33,83,84,85]. However, the precise mechanism by which reduction occurs is not fully understood. In this paper, reduction reaction experiments were undertaken on the synthesised platinum(IV) complexes, **1**–**8**, with the aid of ^1^H-NMR and 1D-^195^Pt-NMR spectroscopy. The method used for the reduction reaction experiments was adapted from the literature without modifications [21]. A total of 10 mM PBS (~7.4 pH) was selected as media, and AsA was used as the biological reducing agent to investigate the reduction property of **1**–**8**. Prior to the reduction reaction experiments, a control was prepared where each metal complex was dissolved with PBS in D_2_O. 1D-^195^Pt-NMR was measured within the regions of −2800 and 400 ppm (30 min per region), at 37 °C. This was undertaken to confirm that the PBS would not interfere with the reduction process of the metal complexes. For the preliminary 1D-^195^Pt-NMR spectra obtained (Appendix A), all complexes were unchanged with no sign of reduction, thus confirming that PBS does not reduce the complexes to their corresponding platinum(II) and platinum(IV) precursor complexes. The approximate time points for the studied complexes at which an estimated 50% reduction had occurred, in the presence of AsA, as represented by T_50%_, are summarised in Table 6.

The reduction property of **1** was monitored for 1 h by ^1^H-NMR spectroscopy at 37 °C (Figure 15). Upon treatment of AsA, the complex started reducing at 5 min as shown in Figure 15. Between 15–20 min, approximately 50% reduction was demonstrated by **1** (Table 5). Accordingly, the resonances of the phen protons (H2 and H9; H4 and H7; H5 and H6; H3 and H8) and the aromatic protons of the 4-CPA ligand (a*,* b*,* c, and d*)* were almost equivalent (Figure 15). At 1 h, the metal complex had almost entirely reduced to its platinum(II) precursor, **PHEN*SS***. A notable observation from the ^1^H-NMR experiments is the movement of resonances, particularly the evident upfield shift from the phen protons and the downfield shift from the 4-CPA protons, as shown in Figure 15.

Subsequently, after the ^1^H-NMR experiments, 1D-^195^Pt-NMR was also carried out centred on the regions of −2800 and 400 ppm, at 37 °C. After 1 h from the final ^1^H-NMR experiment, **1** had reduced entirely to **PHEN*SS*** (Figure 16). As shown in Figure 16, no platinum(IV) resonance was recorded in the 400 ppm region. Additionally, a platinum(II) resonance was recorded in the −2800 ppm region, also validating its complete reduction. The results suggest that **1** could be reduced inside tumour cells and that it could exert antitumor effects via its platinum(II) precursor, and is, therefore, a potential prodrug of **PHEN*SS***.

Furthermore, the reduction property of **2** was also investigated. The reduction kinetics of **2** was similar to **1** (Figure 17). From the ^1^H-NMR spectra shown in Figure 17, the complex began to reduce at 5 min, and new signals originating from its platinum(II) precursor and the aromatic protons of the axial ligand started to appear. About 50% reduction had occurred between 20 and 25 min for **2** (Table 6). After 1 h, **2** had almost reduced and the signals from the prodrug had decreased significantly while the signals from the released or uncoordinated 4-CPA acid became more prominent, further confirming that it had been detached from the platinum centre. Additionally, 1D-^195^Pt-NMR was also carried out centred on the regions of −2800 and 400 ppm, at 37 °C (Figure 18). While no platinum(IV) resonance was recorded at −400 ppm, a platinum(II) resonance was acquired at −2800 ppm, which confirms that **2** was able to entirely reduce back to its platinum(II) precursor, **56ME*SS***.

The reduction properties of the remaining complexes, **3**–**8**, were also investigated and followed a similar propensity to **1** and **2** according to the obtained ^1^H-NMR and 1D-^195^Pt- NMR spectra (**3**: Appendix A; **4**: Appendix A; **5**: Appendix A; **6**: Appendix A; **7**: Appendix A; **8**: Appendix A). Interestingly, **6** and **7** reduced the slowest of the derivatives, wherein T_50%_ occurred at 30 min (Table 6), and traces of platinum(IV) species were still observed in their 1D-^195^Pt-NMR spectra even 1 h after the final ^1^H-NMR experiments (Appendix A). By contrast, **3** and **8** reduced the fastest, wherein T_50%_ was observed to occur between 5 and 10 min (Table 6, and Appendix A). The results confirm that the axial ligands’ halogenation influenced the complexes’ reduction behaviour more than methylation of the heterocyclic ligand. With the exception of **3** and **8**, it can also be deduced that the derivatives with the most electronegative halogens in their axial ligands (Cl, F, and Br) reduced more quickly than the 4-IPA derivatives containing the least electronegative element, I. In summary, complexes **1**–**8** are thus potential prodrugs of their platinum(II) precursors, **PHEN*SS*** and **56ME*SS***.

### 2.5. Growth Inhibition Assays

Complexes **1**–**8** and the halogenated PAA ligands, 4-CPA, 4-FPA, 4-BPA, and 4-IPA, were screened for antiproliferative activity against a panel of cell lines that included HT29 colon, U87 glioblastoma, MCF-7 breast, A2780 ovarian, H460 lung, A431 skin, Du145 prostate, BE2-C neuroblastoma, SJ-G2 glioblastoma, MIA pancreas, the ADDP-resistant ovarian variant, and the non-tumour derived MCF10A breast line. Compound growth inhibition was assessed using the MTT assay after 72 h of treatment. A summary of the determined GI_50_ values is reported in Table 7, including the GI_50_ values of cisplatin, oxaliplatin, and carboplatin, as well as the platinum(II) and platinum(IV) precursors obtained from a previous study [21], for comparison purposes. It is already established that **56ME*SS*** is the most cytotoxic platinum(II) scaffold of its class, demonstrating an average GI_50_ value of 36 ± 10 nM from the panel of cell lines tested, as shown in Table 7. **PHEN*SS*** was of lower potency than **56ME*SS***, and this difference in cytotoxicity is attributed to be the effect of methylation in the 5 and 6 positions of the heterocyclic ligand [16,21,24,73]. Briefly, the structural differences between **PHEN*SS*** and **56ME*SS*** are modest with changes in the pKa of the coordinated nitrogen (N) atoms in their heterocyclic ligands that could modulate the overall positive 2 charge. Whilst at the molecular surface and conformational standpoint 5,6-methylation induces a negligible impact, as denoted by pKa values (i.e., phen = 4.9 versus 5,6-Me_2_phen = 5.6) [86], methylation significantly abrogates N-lone pair localization. As previously reported by Ramírez et al. [86] and others, pKa values are congruent with a variety of other parameters, which include π acceptor potential, charge accepting properties (µ+), and electro-accepting power (ω+) (high π acceptor character = high capacity to accept charge). Comparatively, demethylation confers a significant modulatory effect on these parameters according to the values extrapolated by Ramirez et al. on phen-based cobalt(II) coordination complexes [86].

Furthermore, the cancer-killing properties of **56ME*SS*** and **PHEN*SS*** still surpass that of cisplatin, oxaliplatin, and carboplatin. For example, **56ME*SS*** and **PHEN*SS*** were 1100-fold and 70-fold more reactive, respectively, compared to cisplatin in the cisplatin-resistant HT29 colon cancer cell line (Table 7). In the ADDP-resistant ovarian variant cell line, **56ME*SS*** and **PHEN*SS*** proved to be 2150-fold and almost 150-fold more potent than cisplatin, respectively. Notably, **56ME*SS*** exhibited a GI_50_ value of 4.6 ± 0.4 nM in the Du145 prostate cell line, demonstrating almost 3300-fold and 630-fold increases in potency over carboplatin (15,000 ± 1200 nM) and oxaliplatin (2900 ± 400 nM), respectively.

The platinum(IV) precursor complexes, **PHEN*SS*(IV)(OH)_2_** and **56ME*SS*(IV)(OH)_2_**, still demonstrate potency, but not to the same degree as their platinum(II) precursors. For example, **PHEN*SS*(IV)(OH)_2_** has an average GI_50_ value of 3318 ± 880 nM over the range of cell lines tested, while the corresponding platinum(II) precursor, **PHEN*SS***, has an average GI_50_ value of 434 ± 110 nM, demonstrating an almost 8-fold difference in terms of cytotoxicity. A 4-fold difference was also demonstrated between the average GI_50_ values of **56ME*SS*(IV)(OH)_2_** (151 ± 50 nM) and its platinum(II) precursor, **56ME*SS*** (36 ± 10 nM) in all of the cell lines. This indicates that the addition of hydroxyl or OH moieties in the axial positions to our platinum(IV) scaffold, **[Pt^IV^(H_L_)(A_L_)(OH)_2_]^2+^**, had decreased the overall cytotoxicity of the complexes. Nonetheless, **PHEN*SS*(IV)(OH)_2_** and **56ME*SS*(IV)(OH)_2_** are still significantly more potent than cisplatin, oxaliplatin, and carboplatin (Table 7).

For the new series of platinum(IV) complexes incorporating the halogenated PAA ligands, the most cytotoxic derivatives contained the methylated heterocyclic ring system, 5,6-Me_2_phen (**2**, **4**, **6** and **8**), whereas the least cytotoxic derivatives contained the unmethylated heterocyclic ring system, phen (**1**, **3**, **5**, and **7**), as expected. This observation supports the notion that methylation of the heterocyclic ligand is a critical contributor to tuning the overall biological activity of the complexes. For the axial ligands, 4-CPA, 4-FPA, 4-BPA, and 4-IPA, their cytotoxicity was also tested; however, they did not demonstrate significant cytotoxicity (Table 7). Amongst the platinum(IV) derivatives synthesised, **4**, which contains the 4-FPA ligand, and **6**, which contains the 4-BPA ligand, exhibited the greatest cancer-killing potential with GI_50_ values ranging from 0.7 to 110 nM in the panel of cell lines tested (Table 7). It is also noteworthy that the difference in reactivity between **4** and **6** is marginal, especially in the A2780 ovarian, ADDP-resistant ovarian variant, and H460 lung cancer cell lines, where both displayed the same GI_50_ values. Additionally, **4** and **6** displayed their lowest GI_50_ values in Du145 prostate, MIA pancreas, HT29 colon, and the ADDP-resistant ovarian variant cancer cell lines, as shown in Table 7. Particularly, in the Du145 prostate cancer cell line, **4** exhibited an GI_50_ value 0.7 ± 0.4 nM, which is 6.6-fold more potent than its platinum(II) precursor, **56ME*SS*** (4.6 ± 0.4 nM), and about 1700-fold more potent than cisplatin (1200 ± 100 nM). In the same cell line, **6** (1 ± 0.6 nM) also proved to be 4.6 times more potent than **56ME*SS*** and 1200 times more potent than cisplatin. This suggests that **4** and **6** may be most effective in treating prostate cancer. From the results, it can also be inferred that the conjugation of 4-FPA and 4-BPA to **56ME*SS*(IV)(OH)_2_** enhances the overall activity of the complexes. Furthermore, the third-most cytotoxic platinum(IV) derivative in the group contained the 4-CPA ligand (**2**). 4-CPA has been previously investigated as a safe and effective treatment against estrogen-sensitive breast cancer [63,64,65]. It was our expectation that when 4-CPA is coordinated to **56ME*SS*(IV)(OH)_2_**, a prodrug that demonstrates potent cytotoxicity response towards the estrogenic MCF-7 breast cancer cell line can be generated. Complex **2** displayed a GI_50_ value of 180 ± 77 nM in that cell line, which is almost 2-fold less potent than its platinum(II) precursor, **56ME*SS*** (93 ± 44 nM) (Table 7). The potency of **56ME*SS*** in all of the cell lines remains unsurpassed compared to **2**. Nevertheless, **2** showed selectivity towards the HT29 colon, MIA pancreas, and ADDP-resistant ovarian cancer cell lines instead of the MCF-7 breast cancer cell line. This indicates that the coordination of 4-CPA to **56ME*SS*(IV)(OH)_2_** did not significantly influence overall cytotoxicity unlike 4-FPA and 4-BPA.

The remaining platinum(IV) derivatives (**1**, **3**, **5**, **7**, and **8**) may not be as reactive as **2**, **4**, **6**, or **56ME*SS***, but they are still significantly more potent than cisplatin, oxaliplatin, and carboplatin, and the degree of difference is substantial (Table 7). Intriguingly, the 5,6-Me_2_phen derivative, **8**, which contains 4-IPA, had an average GI_50_ value of 387 ± 50 nM over the cell lines tested, sharing almost the same cytotoxicity values with the phen derivatives **1**, **3**, **5**, and **7**. Evidently, the coordination of 4-IPA to **56ME*SS*(IV)(OH)_2_** did not enhance the overall activity of the complex. Furthermore, the phen derivatives, **1**, **3**, **5**, and **7** were as potent as their platinum(II) precursor, **PHEN*SS***, which suggests that the presence of the axial ligands had little to no impact on the overall activity of the complexes. By contrast, an overall improvement of activity was demonstrated when 4-CPA, 4-FPA, and 4-BPA were coordinated to **56ME*SS*(IV)(OH)_2_**, generating the most cytotoxic platinum(IV) derivatives, **2**, **4**, and **6**. Moreover, **1**–**8** are potent in the ADDP-resistant ovarian variant cell line in comparison to all other cell line populations, indicating that the complexes are not susceptible to the drug resistance mechanisms induced by standard clinical treatment with cisplatin.

### 2.6. ROS Production

Although ROS is required for cancer cell function and progression, it can induce programmed cell death or apoptosis [87]. Modulating the levels of ROS may treat cancer by causing DNA damage, resulting in mitochondrial membrane permeabilization that subsequently signals to the execution caspases (3, 6, and 7) to cleave the DNA repair protein PARP-1 (116 kDa), which produces two fragments (89 and 24 kDa) that reduce the binding affinity and repairing abilities [88,89]. In this study, we report the ROS activity of the studied platinum(IV) complexes, **1**–**8**, together with the individual axial ligands (4-CPA, 4-FPA, 4-BPA, and 4-IPA) in the human cancer cell line, HT29 colon (Figure 19, Table 8, and Appendix A). The ROS potential of their platinum(II) and platinum(IV) precursors in the same cell line is also presented for comparison purposes (Table 8) [21]. The HT29 colon cell line was selected as it is one of the human cancer cell lines where complexes **1**–**8** demonstrated superb antitumor potential (Table 7). HT29 colon cells were stained by DCFH-DA and then treated with the complexes at each specific GI_50_ concentration. Upon ROS production, the DCFH-DA produced dichlorodihydrofluorescein, a fluorescent product. The measured fluorescence was, therefore, proportional to the produced ROS.

Complexes **1**–**8** showed significant production of ROS at 24, 48, and 72 h (*p* < 0.0001), with **4** and **6** producing the greatest ROS of 738 ± 15 RFU and 643 ± 14 RFU at 72 h, respectively (Figure 19, Table 8, and Appendix A). The exceptional surge of ROS produced by **4** and **6** may be causative of cytotoxicity potency, as they elicited GI_50_ values of 6 ± 2 and 7 ± 2 nM, respectively, in the HT29 cancer cell line (Table 7). In comparison, HT29 cells treated with the ligands 4-CPA, 4-FPA, 4-BPA, and 4-IPA exhibited an increase in ROS production relative to the control at 24, 48, and 72 h (Figure 19, Table 8, and Appendix A). The results confirm that the studied platinum(IV) complexes, **1**–**8**, may potentially induce DNA damage through the production of ROS inside cancer cells and subsequently induce apoptosis.

## 3. Materials and Methods

All chemicals and reagents were of analytical grade and used without further purification. The deionised water (d.i.H_2_O) used for the experiments was obtained from a MilliQ^TM^ system (Millipore Australia Pty Ltd., Sydney, NSW, Australia). Potassium tetrachloroplatinate(II), *SS*-DACH, phen, 5,6-Me_2_phen, 4-CPA, 4-FPA, 4-BPA, *N*,*N’*-dicyclohexylcarbodiimide (DCC), N-hydroxysuccinimide (NHS), dimethyl sulfoxide (DMSO), acetonitrile (CH_3_CN), trifluoroacetic acid (TFA), AsA, MF-Millipore^TM^ membrane filters (25 mm, 0.45 µm), and Whatman^®^ (55 mm) cellulose papers were purchased from Sigma-Aldrich, Sydney, NSW, Australia. 4-IPA was purchased from Combi-Blocks Inc, San Diego, CA, USA. Sep-Pak^®^ C_18_-reverse phase columns were procured from Waters Australia Pty Ltd., Sydney, NSW, Australia. Advantec^®^ (55 mm) cellulose filter papers were purchased from Sterlitech Corporation, Auburn, WA, USA. Deuterium oxide 99.9% (D_2_O) and deuterated acetonitrile (CD_3_CN) were purchased from Cambridge Isotope Laboratories, Andover, MA, USA. Methanol (MeOH) was obtained from Honeywell Research Chemicals, NJ, USA. Ethanol (EtOH), diethyl ether (Et_2_O), acetone (C_3_H_6_O), and dichloromethane (DCM) were purchased from ChemSupply, Gillman, SA, Australia. All other chemicals and certain goods were obtained from commercial sources.

### 3.1. Instrumentation

#### 3.1.1. Flash Chromatography

A Biotage Isolera^TM^ One flash chromatography system (Shimadzu, Sydney, NSW, Australia) equipped with a Biotage^®^ Sfär C18 D (Duo 100 Å 30 μm 30 g) (Shimadzu, Sydney, NSW, Australia) was utilised to purify the platinum(IV) complexes. The mobile phase consisted of solvents, A (d.i.H_2_O) and B (MeOH). The samples were dissolved in d.i.H_2_O/MeOH (50:50) and eluted through the column with a 0–30% linear gradient for 40 min with a flow rate of 4 mL·min^−1^, collected within the set wavelengths of 200–400 nm.

#### 3.1.2. High-Performance Liquid Chromatography (HPLC)

An Agilent (Melbourne, VIC, Australia) Technologies 1260 Infinity instrument equipped with a Phenomenex Onyx^TM^ Monolithic C_18_-reverse phase column (100 × 4.6 mm, 5 µm pore size) (Sydney, NSW, Australia) was utilised for the complexes. The mobile phase consisted of solvents, A (0.06% TFA in d.i.H_2_O) and B (0.06% TFA in CH_3_CN/d.i.H_2_O (90:10)). An injection volume of 5 µL was utilised and eluted with a 0–100% linear gradient over 15 min with a flow rate of 1 mL·min^−1^, at the set wavelengths of 214 and 254 nm. An Agilent ZORBAX RX-C_18_ column (100 × 4.6 mm, 3.5 µm pore size) (Sydney, NSW, Australia) was utilised for the NHS esters of 4-CPA, 4-FPA, 4-BPA, and 4-IPA using the aforementioned method.

The stabilities of **1**–**8** were monitored using an Agilent (Melbourne, VIC, Australia) Technologies 1260 Infinity instrument equipped with a Phenomenex Onyx^TM^ Monolithic C_18_-reverse phase column (100 × 4.6 mm, 5 µm pore size) (Sydney, NSW, Australia). The mobile phase consisted of solvents, A (0.06% TFA in d.i.H_2_O) and B (0.06% TFA in CH_3_CN/d.i.H_2_O (90:10)). An injection volume of 5 µL was utilised and eluted with a 0–100% linear gradient over 15 min, at a flow rate of 1 mL·min^−1^, with chromatograms acquired at wavelengths of 214 and 254 nm. Complexes **1**–**8** were dissolved in aqueous phosphate buffered saline (PBS) solution at pH ~7.4 and incubated at 37 °C for 36 h. Additionally, the solubility of **1**–**8** was measured in d.i.H_2_O at room temperature.

For lipophilicity measurements, analytical HPLC was utilised. Elution profiles were acquired on an Agilent (Melbourne, VIC, Australia) Technologies 1260 Infinity instrument equipped with a Phenomenex Onyx^TM^ Monolithic C_18_-reverse phase column (100 × 4.6 mm, 5 µm pore size) (Sydney, NSW, Australia). The mobile phase consisted of solvents, A (0.06% TFA in d.i.H_2_O) and B (0.06% TFA in CH_3_CN/d.i.H_2_O (90:10)). Potassium iodide was used as an external dead volume marker to determine the dead time of the column. Retention times (T_R_) were measured at varying isocratic ratios ranging from 30 to 36% of solvent B at 1 mL·min^−1^ mL. An injection volume of 10 µL was utilised. Capacity factors were determined according to Equation (1):(1)$$k=(T_{R}-T_{0})/T_{0}$$
where k is the capacity factor, T_R_ is the retention time of the analyte, and T_0_ represents the dead time. A minimum of four different mobile compositions were used for each complex to calculate k. A linear plot was generated of log k against the concentration of CH_3_CN in the mobile phase to determine the value of log k_w_ expressed by Equation (2):(2)$$\log k=S\varphi +\log k_{w}$$
where S is the slope, φ is the concentration of the CH_3_CN in the mobile phase, and log k_w_ represents the capacity factor of the complex in 100% d.i.H_2_O. Extrapolation of this linear plot to the *y*-intercept indicates the log k_w_ value.

#### 3.1.3. Nuclear Magnetic Resonance (NMR) Spectroscopy

^1^H-NMR, 2D-COSY, ^1^H-^195^Pt HMQC, 1D-^195^Pt-NMR, and ^19^F-NMR were acquired on a 400 MHz Bruker (Melbourne, VIC, Australia) Avance Spectrometer at 298 K. All samples were prepared to a concentration of 10–20 mM using D_2_O or CD_3_CN. ^1^H-NMR was set to 10 ppm and 16 scans with a spectral width of 8250 Hz and 65,536 data points. 2D-COSY was acquired using a spectral width of 3443 Hz for both the ^1^H nucleus and F1 and F2 dimensions, with 256 and 2048 data points, respectively. ^1^H−^195^Pt-HMQC was carried out using a spectral width of 214,436 Hz and 256 data points for the ^195^Pt nucleus and F1 dimension, and also a spectral width of 4808 Hz with 2048 data points for the ^1^H nucleus and F2 dimension. 1D-^195^Pt-NMR was measured using a spectral width of 85,470 Hz and 674 data points. ^19^F-NMR was acquired at eight scans with a spectral width of 75,188 Hz and 300,742 data points. All recorded resonances were presented as chemical shifts in parts per million (δ ppm) with *J*-coupling constants reported in Hz. For spin multiplicity: s (singlet); d (doublet); t (triplet); q (quartet); and m (multiplet). All spectroscopic data gathered were generated and plotted using TopSpin 4.1.3 analysis software.

The reduction properties of **1**–**8** were monitored using ^1^H-NMR and 1D-^195^Pt-NMR spectroscopy on a 400 MHz Bruker (Melbourne, VIC, Australia) Avance Spectrometer. A sequence of ^1^H-NMR experiments was carried out for 1 h at 37 °C, followed by 1D-^195^Pt-NMR within the regions of −2800 and 400 ppm. An amount of 10 mM PBS (~7.4 pH) was transferred to a vial and reduced to dryness through rotary evaporation. AsA (~1 mg) was combined with the metal complex (~5 mg) and transferred to the vial containing the dried PBS. A total of 600 µL of D_2_O was then added to the vial to dissolve the complex, AsA, and the PBS together. Each reaction was followed at 37 °C until the complete reduction of the complexes.

#### 3.1.4. Ultraviolet-Visible (UV) Spectroscopy

An Agilent (Melbourne, VIC, Australia) Technologies Cary 3500 UV-Vis Multicell Peltier spectrophotometer was utilised to perform the UV spectroscopy experiments. UV spectroscopic experiments were completed at room temperature in the range of 200–400 nm with a 1 cm quartz cuvette. All complexes were prepared in d.i.H_2_O. A stock solution of each complex (1 mM) was prepared and absorption spectra were recorded at a series of different concentrations by titrating 9 × 3 µL aliquots into a cuvette containing d.i.H_2_O (3000 µL). Experiments were repeated in triplicate. All spectra were baseline-corrected by the instrument—a baseline containing d.i.H_2_O was acquired first, and automatically subtracted from each experiment. Average extinction coefficients (ε) were determined with standard deviation and errors based on the generated plot curves.

#### 3.1.5. Circular Dichroism (CD) Spectroscopy

A Jasco (Easton, PA, USA) J-810 CD spectropolarimeter was used to measure the CD spectra of the studied prodrugs. The samples were prepared in d.i.H_2_O in a 1 mm optical glass cuvette or 1 cm quartz cuvette. CD experiments were undertaken at room temperature in the wavelength range of 200–400 nm (30 accumulations) with a bandwidth of 1 nm, data pitch of 0.5 nm, a response time of 1 sec and a 100 nm·min^−1^ scan speed. The flowrate of nitrogen gas was 6 L·min^−1^. The HT (photo-multiplier) level remained below 500 V for all experiments. A CD simulation tool (CDToolX) was used to generate the spectra.

#### 3.1.6. High-Resolution Electrospray Ionization Mass Spectrometry (ESI-MS)

High-resolution ESI-MS experiments were undertaken using a Waters (Sydney, NSW, Australia) SYNAPT G2-Si quadruple time-of-flight (QTOF) HDMS. A stock solution of each complex (1 mM) was prepared in d.i.H_2_O. A total of 5 µL of the stock solution was diluted with 995 µL of d.i.H_2_O to create the sample solution. For the NHS esters of 4-CPA, 4-FPA, 4-BPA, and 4-IPA, CH_3_CN was used to prepare the stock solutions (1 mM). A total of 5 µL of the stock solution for each NHS ester was also diluted with 995 µL of CH_3_CN to create the sample solutions. The wire or capillary where the sample solutions were injected was washed with d.i.H_2_O/CH_3_CN (50:50) before every experiment to avoid cross-contamination.

### 3.2. Chemistry

#### 3.2.1. General Synthesis Route for Precursor Platinum(II) and Platinum(IV) Complexes of Type [Pt^II^(H_L_)(A_L_)]^2+^ and [Pt^IV^(H_L_)(A_L_)(OH)_2_]^2+^

The precursor complexes, **[Pt^II^(H_L_)(A_L_)]^2+^** (**PHEN*SS*** and **56ME*SS***) and **[Pt^IV^(H_L_)(A_L_)(OH)_2_]^2+^** (**PHEN*SS*(IV)(OH)_2_** and **56ME*SS*(IV)(OH)_2_**), were synthesised as described previously, without modifications [21,71].

#### 3.2.2. Synthesis Route for NHS Esters of 4-CPA, 4-FPA, 4-BPA, and 4-IPA

All NHS esters were synthesised as previously described [70,71], but with minor modifications. The acids were reacted with 1 mol eq. of both NHS and DCC in either C_3_H_6_O or DCM. The reaction solution was left stirring in a covered flask at room temperature for 24 h. The precipitated by-product, dicyclohexylurea (DCU), was removed either through vacuum filtration or syringe filtration. The filtrate was collected and rotary evaporated to dryness. All NHS esters were used without further purification.

**NHS-4-CPA** yield: 471 mg; 75%. ^1^H-NMR (400 MHz, CD_3_CN_,_ δ): 7.39 (q, *a* and *b; c* and *d*, 4H, *J =* 8.6 Hz), 4.01 (s, *e*, 2H), 2.79 (s, *f* and *g*, 4H). HPLC, T_R_: 10.7 min. ESI-MS: *calculated* for M^+^ (M = C_12_H_10_ClNO_4_): *m*/*z* = 267.03; *experimental*: *m*/*z* = 267.06.

**NHS-4-FPA** yield: 385 mg; 79%. ^1^H-NMR (400 MHz, CD_3_CN_,_ δ): 7.40 (q, *a* and *b*, 2H, *J* = 5.4 Hz), 7.15 (t, *c* and *d*, 2H, *J* = 8.9 Hz), 3.99 (s, *e*, 2H), 2.79 (s, *f* and *g*, 4H). ^19^F-NMR (400 MHz, CD_3_CN_,_ δ): −116 (m). HPLC, T_R_: 9.74 min. ESI-MS: *calculated* for M^+^ (M = C_12_H_10_FNO_4_): *m*/*z* = 251.06; *experimental*: *m*/*z* = 251.08.

**NHS-4-BPA** yield: 430 mg; 82%. ^1^H-NMR (400 MHz, CD_3_CN_,_ δ): 7.56 (d, *a* and *b*, 2H, *J =* 7.5 Hz), 7.31 (d, *c* and *d*, 2H, *J =* 8.4 Hz), 3.99 (s, *e*, 2H), 2.79 (s, 4H). HPLC, T_R_: 10.9 min. ESI-MS: *calculated* for M^+^ (M = C_12_H_10_BrNO_4_): *m*/*z* = 310.98; *experimental*: *m*/*z* = 310.99.

**NHS-4-IPA** yield: 534 mg; 85%. ^1^H-NMR (400 MHz, CD_3_CN_,_ δ): 7.76 (d, *a* and *b*, 2H, *J =* 8.4 Hz), 7.17 (d, *c* and *d*, 2H, *J =* 8.3 Hz), 3.97 (s, *e*, 2H), 2.79 (s, *f* and *g*, 4H). HPLC, T_R_: 11.3 min. ESI-MS: *calculated* for M^+^ (M = C_12_H_10_INO_4_): *m*/*z* = 358.97; *experimental*: *m*/*z* = 358.96; *experimental*: *m*/*z* = 358.98.

#### 3.2.3. Synthesis Route for Mono-Substituted Platinum(IV) Complexes of Type [Pt^IV^(H_L_)(A_L_)(X)(OH)]^2+^ (**1**–**8**)

**[Pt^IV^(H_L_)(A_L_)(OH)_2_]^2+^** complexes were reacted with 2–3.5 mol eq. of NHS esters of the acids in DMSO (3–4 mL) at room temperature in the dark for 72 h. The reaction solution was taken up in Et_2_O (40 mL) and was vigorously mixed using a plastic pipette. Subsequently, centrifugation was performed to afford a brown oily layer and a colourless supernatant. The colourless supernatant was disposed of, while the brown oily layer was collected and diluted with methanol (3 mL), followed by the addition of excess Et_2_O to afford a beige precipitate. The precipitate (in the form of a pellet) was fragmented and washed with excess C_3_H_6_O, separated again via centrifugation, and left to dry.

**[PHEN*SS*(IV)(4-CPA)(OH)]^2+^** (**PHEN*SS*(IV)-4CPA** or **1**) yield: 142 mg; 96%. ^1^H-NMR (400 MHz, D_2_O_,_ δ): 9.22 (d, H2, 1H, *J =* 5.6 Hz), 9.17 (d, H9, 1H, *J* = 5.5 Hz), 9.03 (t, H4 and H7, 2H, *J* = 7.9 Hz), 8.32 (s, H5 and H6, 2H), 8.20 (m, H3 and H8, 2H), 6.53 (d, *a* and *b*, 2H, *J =* 8.3 Hz), 6.28 (d, *c* and *d*, 2H, *J* = 8.3 Hz), 3.13 (m, H1′ and H2′; *e*, 4H), 2.38 (d, H3′ and H6′ eq., 2H), 1.68 (m, H4′ and H5′ eq.; H3′ and H6′ ax., 4H), and 1.27 (m, H4′ and H5′ ax., 2H). ^1^H−^195^Pt-HMQC (400 MHz, D_2_O_,_ δ): 9.22/523 ppm; 9.17/523 ppm; 8.20/523 ppm; 3.13/523 ppm. HPLC, T_R_: 6.60 min. UV λ_max_ nm (ε/mol^−1^ dm^3^ cm^−1^ ± SD × 10^4^, d.i.H_2_O): 279 (4.50 ± 0.88), 304 (1.27 ± 0.70). CD λ_max_ nm (Δε/M·cm^−1^ × 10^1^, d.i.H_2_O): 261 (−69.2), 288 (+67.7). ESI-MS: *calculated* for [M-H]^+^: *m*/*z* = 674.15; *experimental*: *m*/*z* = 675.15.

**[56ME*SS*(IV)(4-CPA)(OH)]^2+^** (**56ME*SS*(IV)-4CPA** or **2**) yield: 60 mg; 94%. ^1^H-NMR (400 MHz, D_2_O_,_ δ): 9.13 (m, H2 and H9; H4 and H7, 4H), 8.18 (m, H3 and H8, 2H), 6.48 (d, *a* and *b*, 2H, *J =* 8.4 Hz), 6.21 (d, *c* and *d*, 2H, *J* = 8.4 Hz), 3.10 (m, H1′ and H2′; *e*, 4H), 2.86 (d, 2 × CH_3_, 6H, *J =* 5.1 Hz), 2.37 (m, H3′ and H6′ eq., 2H), 1.67 (d, H4′ and H5′ eq.; H3′ and H6′ ax., 4H), and 1.27 (m, H4′ and H5′ ax., 2H). ^1^H−^195^Pt-HMQC (400 MHz, D_2_O_,_ δ): 9.13/531 ppm; 8.18/531 ppm; 3.10/531 ppm. HPLC, T_R_: 7.37 min. UV λ_max_ nm (ε/mol^−1^ dm^3^ cm^−1^ ± SD × 10^4^, d.i.H_2_O): 239 (1.98 ± 1.83), 287 (1.95 ± 1.85), 317 (0.85 ± 0.85). CD λ_max_ nm (Δε/M·cm^−1^ × 10^1^, d.i.H_2_O): 204 (−517), 215 (−351), 218 (−392), 234 (−46.1), 247 (−147), 296 (+78.9). ESI-MS: *calculated* for [M-H]^+^: *m*/*z* = 702.18; *experimental*: *m*/*z* = 703.18.

**[PHEN*SS*(IV)(4-FPA)(OH)]^2+^** (**PHEN*SS*(IV)-4FPA** or **3**) yield: 57 mg; 86%. ^1^H-NMR (400 MHz, D_2_O_,_ δ): 9.21 (d, H2, 1H, *J =* 5.6 Hz), 9.18 (d, H9, 1H, *J =* 5.6 Hz), 9.02 (t, H4 and H7, 2H, *J* = 7.5 Hz), 8.27 (s, H5 and H6, 2H), 8.21 (m, H3 and H8, 2H), 6.35 (m, *a* and *b; c* and *d*, 4H), 3.13 (m, H1′ and H2′; *e*, 4H), 2.37 (d, H3′ and H6′ eq., 2H), 1.67 (m, H4′ and H5′ eq.; H3′ & H6′ ax., 4H), and 1.26 (m, H4′ and H5′ ax., 2H). ^1^H−^195^Pt-HMQC (400 MHz, D_2_O_,_ δ): 9.21/540 ppm; 9.18/540 ppm; 8.21/540 ppm; 3.13/540 ppm. ^19^F-NMR (400 MHz, D_2_O_,_ δ): −116 (m). HPLC, T_R_: 6.33 min. UV λ_max_ nm (ε/M·cm^−1^ ± SD × 10^4^, d.i.H_2_O): 204 (6.88 ± 1.28), 279 (2.55 ± 1.21). CD λ_max_ nm (Δε/M·cm^−1^ × 10^1^, d.i.H_2_O): 204 (−688), 237 (−169), 268 (−72.7). ESI-MS: *calculated* for [M-H]^+^: *m*/*z* = 658.18; *experimental*: *m*/*z* = 658.18.

**[56ME*SS*(IV)(4-FPA)(OH)]^2+^** (**56ME*SS*(IV)-4FPA** or **4**) yield: 66 mg; 85%. ^1^H-NMR (400 MHz, D_2_O_,_ δ): 9.11 (m, H2 and H9; H4 and H7, 4H), 8.17 (m, H3 and H8, 2H), 6.23 (m, *a* and *b*; *c* and *d*, 4H), 3.09 (m, H1′ and H2′; *e*, 4H), 2.82 (d, 2 × CH_3_, 6H, *J =* 1.6 Hz), 2.38 (d, H3′ and H6′ eq., 2H), 1.66 (d, H4′ and H5′ eq.; H3′ and H6′ ax., 4H), and 1.29 (m, H4′ and H5′ ax., 2H). ^1^H−^195^Pt-HMQC (400 MHz, D_2_O_,_ δ): 9.11/528 ppm; 8.17/528 ppm; 3.09/528 ppm. ^19^F-NMR (400 MHz, D_2_O_,_ δ): −116 (m). HPLC, T_R_: 6.83 min. UV λ_max_ nm (ε/M. cm^−1^ ± SD × 10^4^, d.i.H_2_O): 248 (2.66 ± 1.19), 293 (3.30 ± 0.33), 317 (1.34 ± 2.49). CD λ_max_ nm (Δε/M·cm^−1^ × 10^1^, d.i.H_2_O): 204 (−414), 233 (−9.58), 260 (−8.22), 346 (+71.4). ESI-MS: *calculated* for [M-H]^+^: *m*/*z* = 686.21; *experimental*: *m*/*z* = 686.21.

**[PHEN*SS*(IV)(4-BPA)(OH)]^2+^** (**PHEN*SS*(IV)-4BPA** or **5**) yield: 36 mg; 78%. ^1^H-NMR (400 MHz, D_2_O_,_ δ): 9.22 (d, H2, 1H, *J* = 5.5 Hz), 9.17 (d, H9, 1H, *J* = 5.5 Hz), 9.04 (t, H4 and H7, 2H, *J* = 8.6 Hz), 8.34 (q, H5 and H6, 2H, *J* = 9 Hz), 8.20 (m, H3 and H8, 2H), 6.68 (d, *a* and *b*, 2H, *J* = 8.3 Hz), 6.21 (d, *c* and *d*, 2H, *J* = 8.4 Hz), 3.10 (m, H1′ and H2′; *e*, 4H), 2.38 (d, H3′ and H6′ eq., 2H), 1.69 (m, H4′ and H5′ eq.; H3′ and H6′ ax., 4H), and 1.27 (m, H4′ and H5′ ax., 2H). ^1^H−^195^Pt-HMQC (400 MHz, D_2_O_,_ δ): 9.22/540 ppm; 9.17/540 ppm; 8.20/540 ppm. HPLC, T_R_: 6.82 min. UV λ_max_ nm (ε/M·cm^−1^ ± SD × 10^4^, d.i.H_2_O): 279 (2.77 ± 0.97), 307 (0.74 ± 0.89). CD λ_max_ nm (Δε/M·cm^−1^ × 10^1^, d.i.H_2_O): 205 (−486), 253 (+7.35), 289 (+83.3). ESI-MS: *calculated* for [M-H]^+^: *m*/*z* = 718.10; *experimental*: *m*/*z* = 719.09.

**[56ME*SS*(IV)(4-BPA)(OH)]^2+^** (**56ME*SS*(IV)-4BPA** or **6**) yield: 31 mg; 72%. ^1^H-NMR (400 MHz, D_2_O_,_ δ): 9.12 (m, H2 and H9; H4 and H7, 4H), 8.17 (m, H3 and H8, 2H), 6.63 (d, *a* and *b*, 2H, *J =* 8.2 Hz), 6.16 (d, *c* and *d*, 2H, *J* = 8.2 Hz), 3.09 (m, H1′ and H2′; *e*, 4H), 2.87 (d, 2 × CH_3_, 6H, *J =* 6.2 Hz), 2.38 (m, H3′ and H6′ eq., 2H), 1.66 (m, H4′ and H5′ eq.; H3′ and H6′ ax., 4H), and 1.28 (m, H4′ and H5′ ax., 2H). ^1^H−^195^Pt-HMQC (400 MHz, D_2_O_,_ δ): 9.12/528 ppm; 8.17/528 ppm. HPLC, T_R_: 7.46 min. UV λ_max_ nm (ε/M·cm^−1^ ± SD × 10^4^, d.i.H_2_O): 248 (1.76 ± 2.06), 291 (2.39 ± 0.31), 317 (0.66 ± 3.22). CD λ_max_ nm (Δε/M·cm^−1^ × 10^1^, d.i.H_2_O): 206 (−522), 218 (−332), 238 (−79.9), 246 (−128), 259 (−36.3), 280 (−14.7). ESI-MS: *calculated* for [M-H]^+^: *m*/*z* = 746.13; *experimental*: *m*/*z* = 747.13.

**[PHEN*SS*(IV)(4-IPA)(OH)]^2+^** (**PHEN*SS*(IV)-4IPA** or **7**) yield: 45 mg; 78%. ^1^H-NMR (400 MHz, D_2_O_,_ δ): 9.20 (d, H2, 1H, *J =* 5.6 Hz), 9.17 (d, H9, 1H, *J =* 5.5 Hz), 9.06 (q, H4 and H7, 2H, *J =* 8.4 Hz), 8.39 (q, H5 and H6, 2H, *J =* 8.9 Hz), 8.20 (m, H3 and H8, 2H), 6.88 (d, *a* and *b*, 2H, *J =* 8.2 Hz), 6.08 (d, *c* and *d*, 2H, *J* = 8.2 Hz), 3.11 (m, H1′ and H2′; *e*, 4H), 2.38 (d, H3′ and H6′ eq., 2H), 1.69 (m, H4′ and H5′ eq.; H3′ and H6′ ax., 4H), and 1.28 (m, H4′ and H5′ ax., 2H). ^1^H−^195^Pt-HMQC (400 MHz, D_2_O_,_ δ): 9.20/540 ppm; 8.39/540 ppm; 3.11/540 ppm. HPLC, T_R_: 6.84 min. UV λ_max_ nm (ε/M. cm^−1^ ± SD × 10^4^, d.i.H_2_O): 280 (3.19 ± 0.54), 306 (0.92 ± 0.82). CD λ_max_ nm (Δε/M·cm^−1^ × 10^1^, d.i.H_2_O): 202 (−565), 217 (−290), 232 (−188), 238 (−248), 270 (−7.91). ESI-MS: *calculated* for [M-H]^+^: *m*/*z* = 766.08; *experimental*: *m*/*z* = 766.08.

**[56ME*SS*(IV)(4-IPA)(OH)]^2+^** (**56ME*SS*(IV)-4IPA** or **8**) yield: 38 mg; 76%. ^1^H-NMR (400 MHz, D_2_O_,_ δ): 9.13 (m, H2 and H9; H4 and H7, 4H), 8.17 (m, H3 and H8, 2H), 6.85 (d, *a* and *b*, 2H, *J =* 8.1 Hz), 6.04 (d, *c* and *d*, 2H, *J* = 8.1 Hz), 3.07 (m, H1′ and H2′; *e*, 4H), 2.91 (d, 2 × CH_3_, 6H, *J =* 7.8 Hz), 2.38 (d, H3′ and H6′ eq., 2H), 1.66 (m, H4′ and H5′ eq.; H3′ and H6′ ax., 4H), and 1.29 (m, H4′ and H5′ ax., 2H). ^1^H−^195^Pt-HMQC (400 MHz, D_2_O_,_ δ): 9.13/530 ppm; 8.17/530 ppm. HPLC, T_R_: 7.62 min. UV λ_max_ nm (ε/M. cm^−1^ ± SD × 10^4^, d.i.H_2_O): 230 (5.29 ± 1.72), 290 (2.76 ± 0.99), 317 (0.76 ± 0.88). CD λ_max_ nm (Δε/M·cm^−1^ × 10^1^, d.i.H_2_O): 206 (−558), 237 (−94), 246 (−193), 276 (−28.9). ESI-MS: *calculated* for [M-H]^+^: *m*/*z* = 794.12; *experimental*: *m*/*z* = 794.12.

### 3.3. Biological Investigations

#### 3.3.1. Cell Growth Assays

Cell growth assays were performed at the Calvary Mater Newcastle Hospital, Waratah, Newcastle, NSW 2298, Australia. The cell lines tested were: HT29 colon, U87 glioblastoma, MCF-7 breast, A2780 ovarian, H460 lung, A431 skin, Du145 prostate, BE2-C neuroblastoma, SJ-G2 glioblastoma, MIA pancreas, the ADDP-resistant ovarian variant, and the non-tumour derived MCF10A breast line. In addition to **1**–**8**, 4-CPA, 4-FPA, 4-BPA, and 4-IPA were also tested for reference and comparison. All test agents were prepared in DMSO (30 mM stock solutions) and stored at −20 °C until use. All cell lines were cultured in a humidified atmosphere of 5% carbon dioxide at 37 °C. The cancer cell lines were maintained in Dulbecco’s modified Eagle’s medium (DMEM) (Trace Biosciences, Sydney, NSW, Australia) supplemented with 10% foetal bovine serum, 10 mM sodium bicarbonate penicillin (100 IU mL^−1^), streptomycin (100 µg mL^−1^), and glutamine (4 mM). The non-cancer MCF10A cell line was cultured in DMEM:F12 (1:1) cell culture media, 5% heat inactivated horse serum, supplemented with penicillin (50 IU mL^−1^), streptomycin (50 µg mL^−1^), 20 mM 4-(2-hydroxyethyl)-1-piperazineethanesulfonic acid (HEPES), L-glutamine (2 mM), epidermal growth factor (20 ng mL^−1^), hydrocortisone (500 ng mL^−1^), cholera toxin (100 ng mL^−1^), and insulin (10 μg mL^−1^). Cytotoxicity was determined by plating cells in duplicate in 100 mL medium at a density of 2500–4000 cells per well in 96 well plates. On day 0 (24 h after plating) when the cells were in logarithmic growth, 100 μL of medium with or without the test agent was added to each well. After 72 h, the GI_50_ was evaluated using the MTT (3-[4,5-dimethylthiazol-2-yl]-2,5-diphenyltetrazolium bromide) assay, and absorbance was read at 540 nm. An eight-point dose response curve was produced from which the GI_50_ value was calculated, representing the drug concentration at which cell growth was inhibited by 50% based on the difference between the optical density values on day 0 and those at the end of drug exposure [90]. The GI_50_ values presented for all precursor platinum(II) and platinum(IV) scaffolds, including cisplatin, oxaliplatin, and carboplatin were obtained from a previous study using the aforementioned method [21].

#### 3.3.2. ROS Detection Assay

To investigate the presence of ROS in treated cells, a DCFDA/H2DCFDA-cellular ROS Assay Kit (Abcam, Cambridge, MA, USA) was used, as previously described [21,91,92,93]. A total of 25,000 cells/mL of HT29 cells in DMEM were seeded in 96-well plates. Cells were washed with 1X kit buffer, and then stained with 25 μM 2′,7′-dichlorofluorescein diacetate (DCFH-DA) and incubated for 45 min. DCFH-DA was then removed, and cells were then re-washed with 1X kit buffer, after which phenol red free media was added. Cells were then treated with a GI_50_ drug concentration for each complex. The plates were directly scanned to measure fluorescence (relative fluorescence units (RFU)) at different time points using the Glo-Max^®^-Multimode microplate reader (Promega Corporation, Alexandra, VIC, Australia) at an excitation/emission of 485/535 nm. To generate the positive control (20 μM *tert*-butyl hydroperoxide (TBHP)), cells were washed with 1X kit buffer, and stained with DCFDA (25 μM) for 45 min; this was removed and TBHP was added in phenol red free media, and the resulting solution was scanned as described above.

## 4. Conclusions

The structure and purity of eight novel platinum(IV) complexes (**1**–**8**) incorporating halogenated PAA derivatives, 4-CPA, 4-FPA, 4-BPA, and 4-IPA, were confirmed by HPLC, NMR, UV, CD, and ESI-MS. NMR was used to monitor the reduction behaviour of the complexes in the presence of AsA. HPLC provided insights into the stability and lipophilicity of the complexes. All complexes proved to be soluble in aqueous solution, and surprisingly stable for 36 h at 37 °C in 10 mM PBS (~7.4 pH). As for the lipophilicity results, the phen derivatives were less lipophilic than the 5,6-Me_2_phen derivatives. The order of increasing lipophilicity of the complexes also parallels the order of increasing size of the halogens incorporated into the axial ligands (F < Cl < Br < I). No correlation was demonstrated between lipophilicity and cytotoxicity of the complexes. The in vitro results also confirmed that **1**–**8** surpassed the potency of cisplatin and its derivatives in all of the cell lines tested. Notably, **4** and **6** demonstrated strong selectivity towards the cell lines HT29 colon, Du145 prostrate, MIA pancreas, ADDP-resistant ovarian, and MCF10A, with GI_50_ values ranging between 0.7 and 11 nM. In the ADDP-resistant ovarian variant cell line, **4** and **6** were almost 4700-fold more potent than cisplatin. There was some correlation between ROS potential and cytotoxicity in the HT29 colon cell line. The most biologically active complex, **4**, demonstrated the most significant production of ROS, while the least cytotoxic complex, **3**, demonstrated the lowest observable production of ROS. In summary, the findings of this study are a testament that modifying certain structural features is essential in tuning the biological activity of novel therapeutic molecules. Furthermore, stability and reduction experiments on the studied complexes in human blood plasma or serum using NMR are scheduled in the near future.

## 5. Patents

This work is part of Australian Provisional Patent Application 2022900110, Platinum(IV) complexes, February 2022, Western Sydney University, Sydney, Australia.

## Figures and Tables

**Figure 1 molecules-27-07120-f001:**
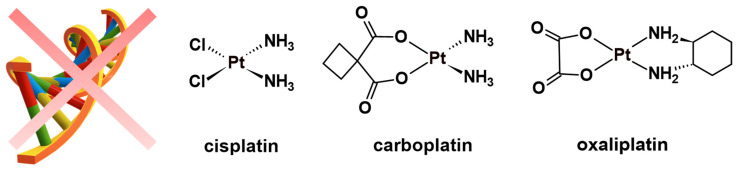
Chemical structures of classical DNA-binding chemotherapeutics.

**Figure 2 molecules-27-07120-f002:**
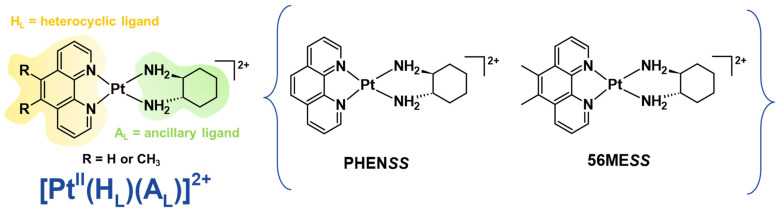
General structures of cytotoxic non-DNA coordinating platinum(II) complexes, **PHEN*SS*** and its analogue, **56ME*SS***.

**Figure 3 molecules-27-07120-f003:**
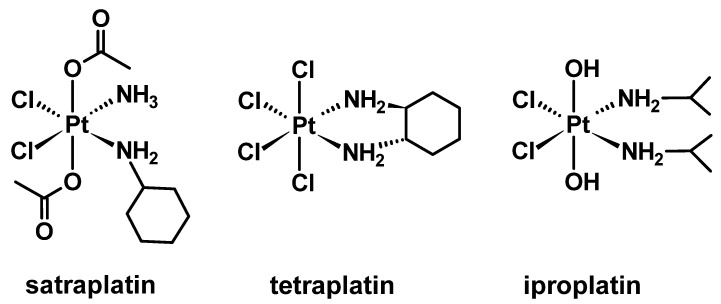
Platinum(IV) complexes that were clinically investigated but failed clinical trials.

**Figure 4 molecules-27-07120-f004:**
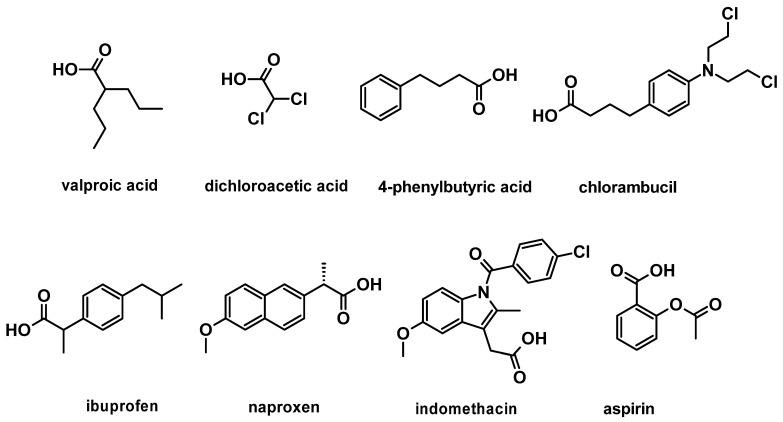
Structures of previously studied bioactive molecules as axial ligands to platinum(IV).

**Figure 5 molecules-27-07120-f005:**
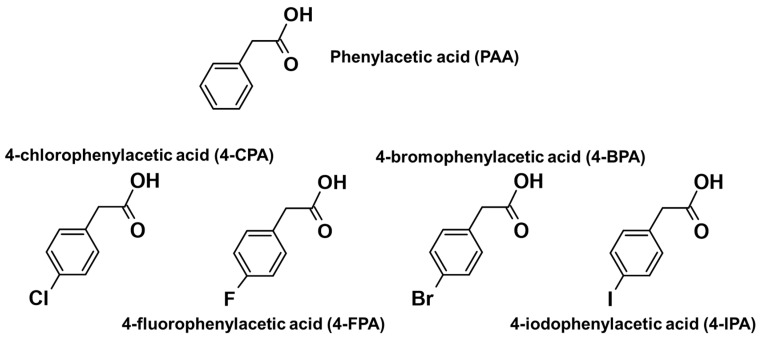
Structures of PAA and its halogenated derivatives.

**Figure 6 molecules-27-07120-f006:**
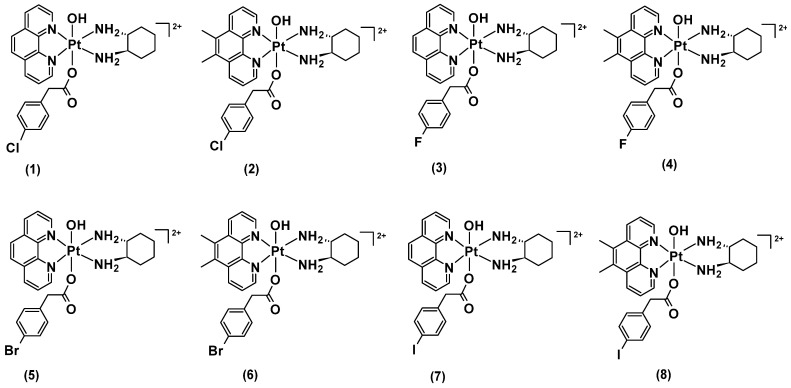
Chemical structures of the studied platinum(IV) complexes (**1**–**8**), incorporating the halogenated PAA derivatives, 4-CPA, 4-FPA, 4-BPA, and 4-IPA. Counter-ions were omitted for clarity.

**Figure 7 molecules-27-07120-f007:**
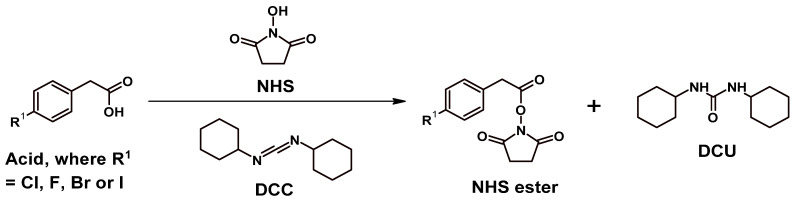
General synthetic pathway for NHS esters.

**Figure 8 molecules-27-07120-f008:**
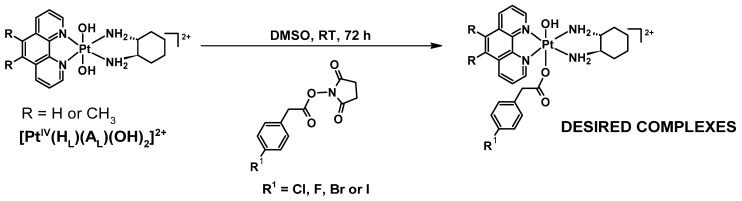
Synthesis route to achieve the desired platinum(IV) complexes incorporating the halogenated PAA derivatives, 4-CPA, 4-FPA, 4-BPA, and 4-IPA.

**Figure 9 molecules-27-07120-f009:**
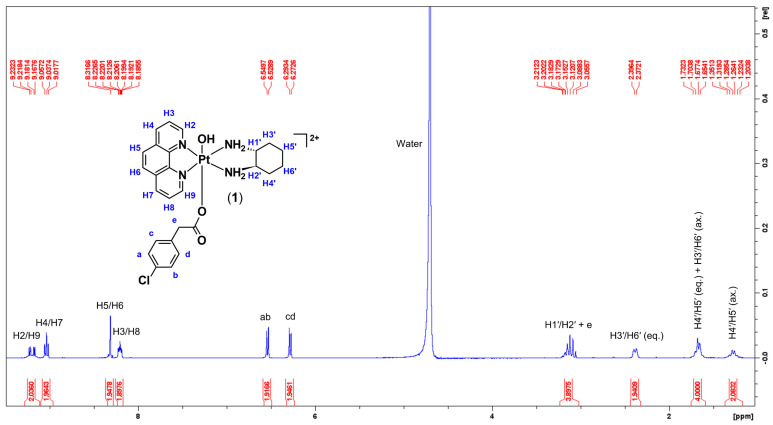
^1^H−NMR spectrum of **1** in D_2_O obtained at 298 K, with proton assignment. Inset: structure of **1** with proton labelling system.

**Figure 10 molecules-27-07120-f010:**
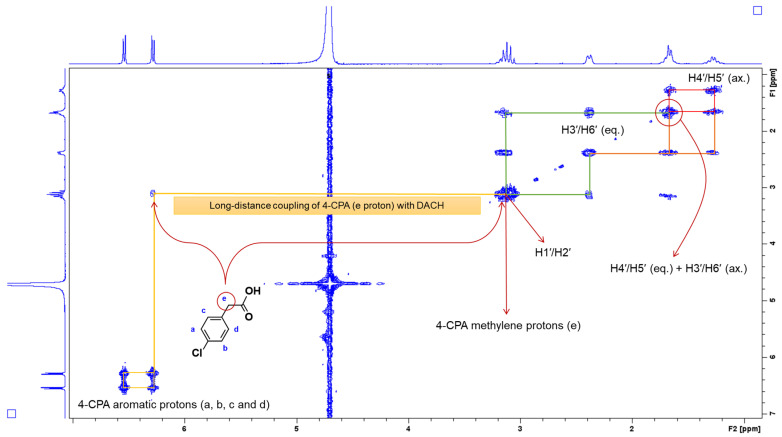
Expanded 2D-COSY spectra of **1** highlighting the correlation between the methylene protons of 4-CPA, e and the DACH protons, H1′ and H2′. Inset: structure of uncoordinated 4-CPA with proton labelling system.

**Figure 11 molecules-27-07120-f011:**
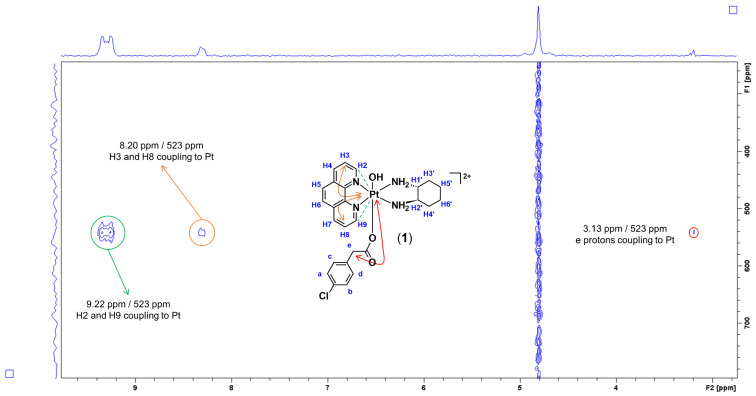
Expanded ^1^H−^195^Pt-HMQC spectrum of **1** in D_2_O, obtained at 298 K highlighting the coupling between protons and the platinum centre. Inset: structure of **1** with proton labelling system.

**Figure 12 molecules-27-07120-f012:**
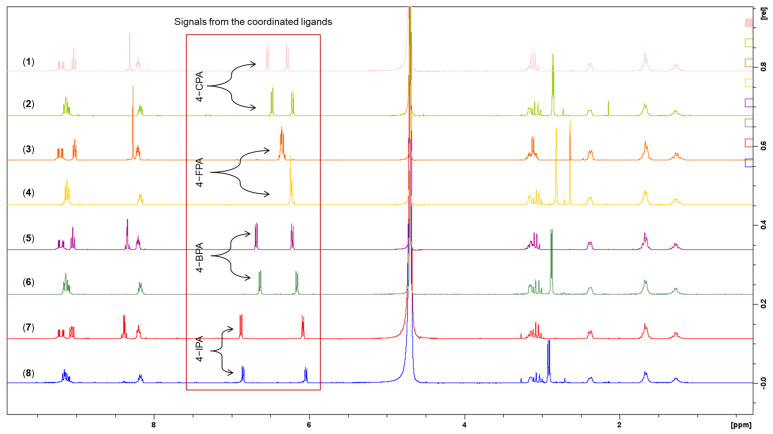
Stacked ^1^H−NMR spectra of **1**–**8** highlighting the differences in splitting from the coordinated PAA ligands, in the region of 6–7 ppm.

**Figure 13 molecules-27-07120-f013:**
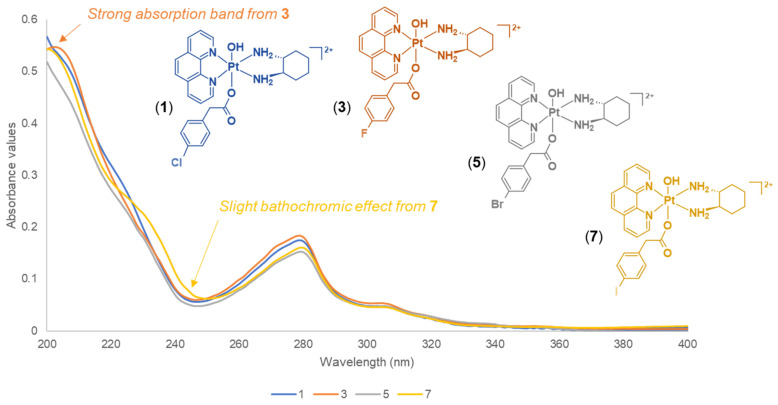
Stacked UV spectra of phen derivatives, **1**, **3**, **5**, and **7**, obtained at 298 K, showing UV absorptions at different wavelengths. Inset: colour-coded structures of **1**, **3**, **5** and **7**.

**Figure 14 molecules-27-07120-f014:**
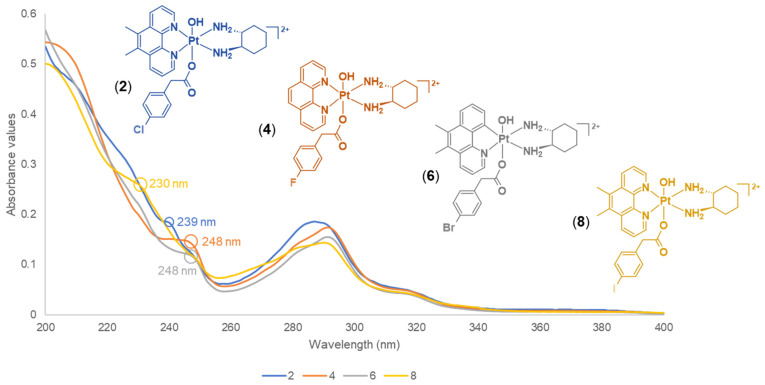
Stacked UV spectra of 5,6-Me_2_phen derivatives, **2**, **4**, **6** and **8** obtained at 298 K, showing UV absorptions at different wavelengths. Inset: colour-coded structures of **2**, **4**, **6** and **8**.

**Figure 15 molecules-27-07120-f015:**
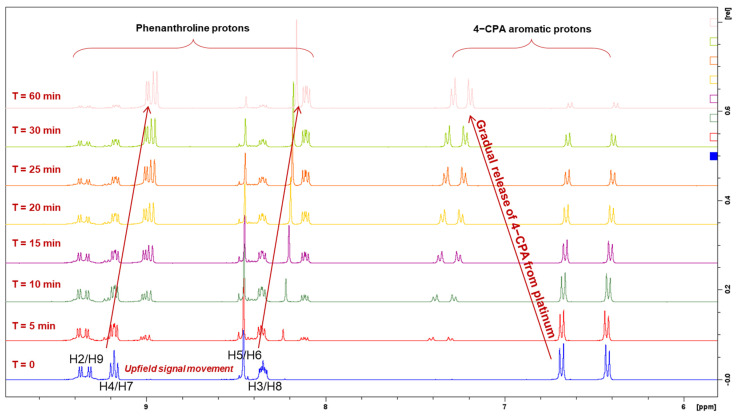
Zoomed ^1^H−NMR spectra of **1** with PBS and AsA in D_2_O at 37 °C, in different time intervals, highlighting the movement of resonances from the phen protons and the aromatic protons of the 4−CPA ligand as indicated by the red arrows. **T** represents time in min.

**Figure 16 molecules-27-07120-f016:**
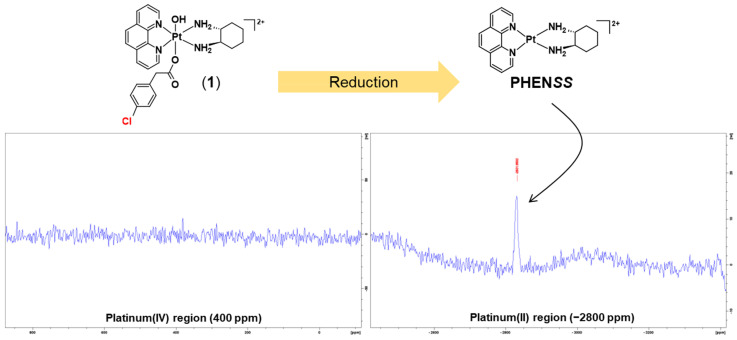
1D−^195^Pt−NMR spectra of **1** with PBS and AsA in D_2_O, within the regions of −2800 and 400 ppm at 37 °C, highlighting its complete reduction after 1 h after the final ^1^H−NMR experiment.

**Figure 17 molecules-27-07120-f017:**
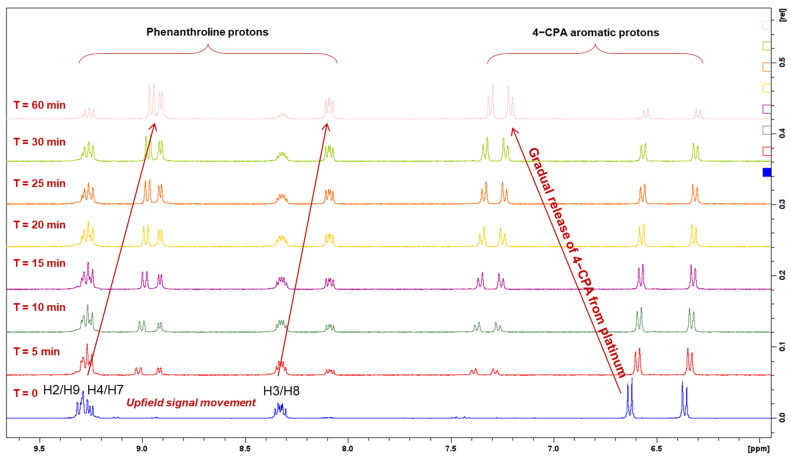
Zoomed ^1^H−NMR spectra of **2** with PBS and AsA in D_2_O at 37 °C, in different time intervals, highlighting the movement of resonances from the phen protons and the aromatic protons of the 4−CPA ligand. **T** represents time in min.

**Figure 18 molecules-27-07120-f018:**
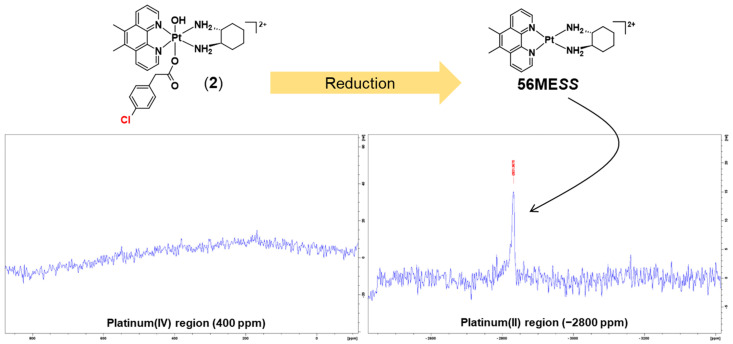
1D−^195^Pt−NMR spectra of **2** with PBS and AsA in D_2_O, within the regions of −2800 and 400 ppm at 37 °C, highlighting its complete reduction after 1 h from the final ^1^H−NMR experiment.

**Figure 19 molecules-27-07120-f019:**
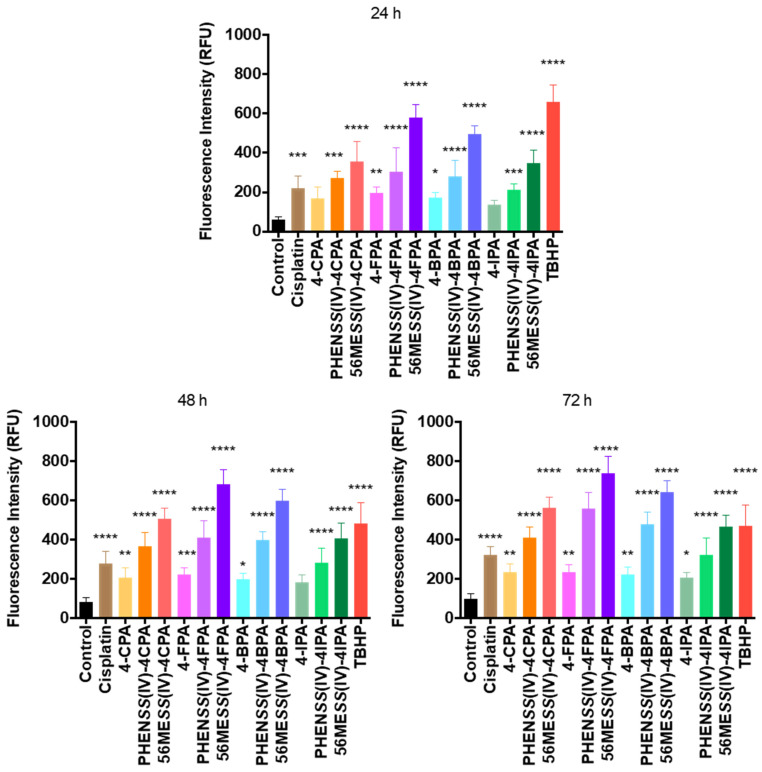
ROS production upon treatment with **1**–**8**, 4-CPA, 4-FPA, 4-BPA, 4-IPA, and cisplatin in HT29 colon cells at 24, 48, and 72 h. **1** = **PHEN*SS*(IV)-4CPA**; **2** = **56ME*SS*(IV)-4CPA**; **3** = **PHEN*SS*(IV)-4FPA**; **4** = **56ME*SS*(IV)-4FPA**; **5** = **PHEN*SS*(IV)-4BPA**; **6** = **56ME*SS*(IV)-4BPA**; **7** = **PHEN*SS*(IV)-4IPA**; **8** = **56ME*SS*(IV)-4IPA**. **** indicates *p* < 0.0001 compared with the control. *** indicates *p* < 0.001 compared with the control. ** indicates *p* < 0.01, compared with the control. * indicates *p* < 0.05 compared with the control. Data points denote mean ± SEM. *n* = 3 from three independent experiments where samples were run in triplicate.

**Table 1 molecules-27-07120-t001:** A summary of experimental yields (%), HPLC peak areas (%), T_R_ (min), and mass-to-charge ratios (*m*/*z*) of the studied complexes.

Platinum(IV) Complexes	Yields (%)	T_R_ (min)	HPLC Peak Areas (%)	Mass-to-Charge Ratios (*m*/*z*)	
				Calc.	Exp.
**1**	96.0	6.60	97	674.15	675.15
**2**	94.0	7.37	98	702.18	703.18
**3**	86.0	6.33	99	658.18	658.18
**4**	85.0	6.83	99	686.21	686.21
**5**	78.0	6.82	99	718.10	719.10
**6**	72.0	7.46	99	746.13	747.13
**7**	78.0	6.84	99	766.08	766.08
**8**	76.0	7.62	99	794.12	794.12

Calc. = calculated; Exp. = experimental.

**Table 2 molecules-27-07120-t002:** Summary of the ^1^H-NMR, ^1^H-^195^Pt-HMQC, and ^19^F-NMR data of **1**–**8** including chemical shifts (δ ppm), multiplicity, integration, and *J*-coupling constants (Hz). Since D_2_O was used in the experiments, no amine resonances were observed due to proton exchange.

Labels	1	2	3	4	5	6	7	8
H2	9.22 (d, 1H, *J =* 5.6 Hz)	9.13 (m, 4H) (overlapping signals from H2, H9, H4, and H7)	9.21 (d, 1H, *J =* 5.6 Hz)	9.11 (m, 4H) (overlapping signals from H2, H9, H4, and H7)	9.22 (d, 1H, *J =* 5.5 Hz)	9.12 (m, 4H) (overlapping signals from H2, H9, H4, and H7)	9.20 (d, 1H, *J =* 5.6 Hz)	9.13 (m, 4H) (overlapping signals from H2, H9, H4, and H7)
H9	9.17 (d, 1H, *J =* 5.5 Hz)		9.18 (d, 1H, *J =* 5.6 Hz)		9.17 (d, 1H, *J =* 5.5 Hz)		9.17 (d, 1H, *J* = 5.5 Hz)	
H4	9.03 (t, 2H, *J* = 7.9 Hz)		9.02 (t, 2H, *J* = 7.5 Hz)		9.04 (t, 2H, *J* = 8.6 Hz)		9.06 (q, 2H, *J =* 8.4 Hz)	
H7								
H5	8.32 (s, 2H)	-	8.27 (s, 2H)	-	8.34 (q, 2H, *J* = 9 Hz)	-	8.39 (q, 2H, *J =* 8.9 Hz)	-
H6								
H3	8.20 (m, 2H)	8.18 (m, 2H)	8.21 (m, 2H)	8.17 (m, 2H)	8.20 (m, 2H)	8.17 (m, 2H)	8.20 (m, 2H)	8.17 (m, 2H)
H8								
a and b	6.53 (d, 2H, *J =* 8.3 Hz)	6.48 (d, 2H, *J =* 8.4 Hz)	6.35 (m, 4H) (overlapping signals from a and b, and c and d because of F electronegativity)	6.23 (m, 4H) (overlapping signals from a and b, and c and d because of F electronegativity)	6.68 (d, 2H, *J =* 8.3 Hz)	6.63 (d, 2H, *J =* 8.2 Hz)	6.88 (d, 2H, *J* = 8.2 Hz)	6.85 (d, 2H, *J* = 8.1 Hz)
c and d	6.28 (d, 2H, *J* = 8.3 Hz)	6.21 (d, 2H, *J* = 8.4 Hz)			6.21 (d, 2H, *J* = 8.4 Hz)	6.16 (d, 2H, *J* = 8.2 Hz)	6.08 (d, 2H, *J* = 8.2 Hz)	6.04 (d, 2H, *J* = 8.1 Hz)
2 × CH_3_ (5, 6 positions)	-	2.86 (d, 6H, *J =* 5.1 Hz)	-	2.82 (d, 6H, *J =* 1.6 Hz)	-	2.87 (d, 6H, *J =* 6.2 Hz)	-	2.91 (d, 6H, *J =* 7.8 Hz)
H1′ and H2′; e	3.13 (m, 4H)	3.10 (m, 4H)	3.13 (m, 4H)	3.09 (m, 4H)	3.10 (m, 4H)	3.09 (m, 4H)	3.11 (m, 4H)	3.07 (m, 4H)
H3′ and H6′ eq.	2.38 (d, 2H)	2.37 (m, 2H)	2.37 (d, 2H)	2.38 (d, 2H)	2.38 (d, 2H)	2.38 (d, 2H)	2.38 (d, 2H)	2.38 (d, 2H)
H4′ and H5′ eq.; H3′ and H6′ ax.	1.68 (m, 4H)	1.67 (d, 4H)	1.67 (m, 4H)	1.66 (d, 4H)	1.69 (m, 4H)	1.66 (m, 4H)	1.69 (m, 4H)	1.66 (m, 4H)
H4′ and H5′ ax.	1.27 (m, 2H)	1.27 (m, 2H)	1.26 (m, 2H)	1.29 (m, 2H)	1.27 (m, 2H)	1.28 (m, 2H)	1.28 (m, 2H)	1.29 (m, 2H)
^1^H/^195^Pt	9.22, 9.17, 8.20, 3.13/523	9.13, 8.18, 3.10/531	9.21, 9.18, 8.21, 3.13/540	9.11, 8.17, 3.09/528	9.22, 9.17, 8.20/540	9.12, 8.17/528	9.20, 8.39, 3.11/540	9.13, 8.17/530
^19^F	-	-	−116 (m)	−116 (m)	-	-	-	-

**Table 3 molecules-27-07120-t003:** Band maxima in the UV and CD spectra of **1**–**8**.

Platinum(IV) Complexes	UV/λ_max_ nm (ε/M·cm^−1^ ± SD × 10^4^)	CD/λ_max_ nm (Δε/M·cm^−1^ × 10^1^)
**1**	279 (4.50 ± 0.88), 304 (1.27 ± 0.70)	261 (−69.2), 288 (+67.7)
**2**	239 (1.98 ± 1.83), 287 (1.95 ± 1.85), 317 (0.85 ± 0.85)	204 (−517), 215 (−351), 218 (−392), 234 (−46.1), 247 (−147), 296 (+78.9)
**3**	204 (6.88 ± 1.28), 279 (2.55 ± 1.21)	204 (−688), 237 (−169), 268 (−72.7)
**4**	248 (2.66 ± 1.19), 293 (3.30 ± 0.33), 317 (1.34 ± 2.49)	204 (−414), 233 (−9.58), 260 (−8.22), 346 (+71.4)
**5**	279 (2.77 ± 0.97), 307 (0.74 ± 0.89)	205 (−486), 253 (+7.35), 289 (+83.3)
**6**	248 (1.76 ± 2.06), 291 (2.39 ± 0.31), 317 (0.66 ± 3.22)	206 (−522), 218 (−332), 238 (−79.9), 246 (−128), 259 (−36.3), 280 (−14.7)
**7**	280 (3.19 ± 0.54), 306 (0.92 ± 0.82)	202 (−565), 217 (−290), 232 (−188), 238 (−248), 270 (−7.91)
**8**	230 (5.29 ± 1.72), 290 (2.76 ± 0.99), 317 (0.76 ± 0.88)	206 (−558), 237 (−94), 246 (−193), 276 (−28.9)

**Table 4 molecules-27-07120-t004:** Solubility of platinum(IV) complexes (**1**–**8**) in d.i.H_2_O at room temperature, expressed in mol/L and mg/mL.

Complexes (H_L_ = phen)	Axial Ligands	Solubility		Complexes (H_L_ = 5,6-Me_2_phen)	Axial Ligands	Solubility	
		mol/L	mg/mL			mol/L	mg/mL
**1**	4-CPA	7.7 × 10^−3^	6.2	**2**	4-CPA	3.1 × 10^−2^	26
**3**	4-FPA	2.1 × 10^−2^	17	**4**	4-FPA	4.1 × 10^−2^	34
**5**	4-BPA	6.1 × 10^−3^	5.1	**6**	4-BPA	7.3 × 10^−3^	6.4
**7**	4-IPA	6.2 × 10^−3^	5.5	**8**	4-IPA	6.9 × 10^−3^	6.4

**Table 5 molecules-27-07120-t005:** A summary of log k_w_ values of **1**–**8**.

Complexes (H_L_ = phen)	Axial Ligands	log k_w_	Complexes (H_L_ = 5,6-Me_2_phen)	Axial Ligands	log k_w_
**1**	4-CPA	0.75	**2**	4-CPA	0.94
**3**	4-FPA	0.25	**4**	4-FPA	0.80
**5**	4-BPA	0.87	**6**	4-BPA	1.07
**7**	4-IPA	0.90	**8**	4-IPA	1.19

**Table 6 molecules-27-07120-t006:** A summary of approximate time points in min for **1**–**8** at which 50% reduction occurs as represented by T_50%_.

Platinum(IV) Complexes	T_50%_ (min)
**1**	15–20
**2**	20–25
**3**	5–10
**4**	20–25
**5**	15–20
**6**	30
**7**	30
**8**	5–10

**Table 7 molecules-27-07120-t007:** A summary of the GI_50_ values (nM) of **1**–**8**, 4-CPA, 4-FPA, 4-BPA, and 4-IPA in multiple cell lines. The data presented for cisplatin, oxaliplatin, carboplatin, and precursor platinum(II) and platinum(IV) precursors were obtained from a previous study [21]. nd = not determined.

GI_50_ Values (nM)													
*Platinum(IV) Prodrugs*	HT29	U87	MCF-7	A2780	H460	A431	Du145	BE2-C	SJ-G2	MIA	MCF10A	ADDP	Average GI_50_
**1**	150 ± 64	740 ± 230	860 ± 180	280 ± 67	380 ± 81	560 ± 73	140 ± 43	490 ± 93	420 ± 92	270 ± 42	290 ± 90	290 ± 50	406 ± 90
**2**	22 ± 9.2	79 ± 30	180 ± 77	43 ± 10	200 ± 170	41 ± 8	33 ± 21	110 ± 18	100 ± 45	28 ± 5.5	35 ± 7.8	26 ± 4.5	75 ± 30
**3**	210 ± 30	1000 ± 150	740 ± 60	340 ± 70	320 ± 30	450 ± 60	100 ± 17	810 ± 200	337 ± 60	230 ± 30	320 ± 10	250 ± 10	426 ± 60
**4**	6 ± 2	37 ± 5	23 ± 5	20 ± 4	10 ± 3	14 ± 2	0.7 ± 0.4	110 ± 20	30 ± 8	5 ± 2	11 ± 3	6 ± 2	23 ± 4
**5**	190 ± 20	1100 ± 230	850 ± 240	380 ± 50	320 ± 20	450 ± 10	110 ± 11	960 ± 120	350 ± 20	200 ± 20	320 ± 20	250 ± 20	457 ± 70
**6**	7 ± 2	35 ± 5	21 ± 5	20 ± 3	10 ± 2	19 ± 0.9	1 ± 0.6	100 ± 20	23 ± 5	7 ± 2	10 ± 2	6 ± 2	22 ± 4
**7**	220 ± 33	1010 ± 95	490 ± 134	280 ± 42	380 ± 59	500 ± 180	110 ± 25	680 ± 38	590 ± 23	360 ± 40	280 ± 30	280 ± 24	432 ± 60
**8**	178 ± 31	900 ± 58	420 ± 62	260 ± 45	390 ± 33	430 ± 160	100 ± 21	650 ± 81	520 ± 29	300 ± 55	260 ± 22	230 ± 15	387 ± 50
** *Platinum(II) Precursors* **													
**PHEN*SS***	160 ± 45	980 ± 270	1500 ± 500	230 ± 30	360 ± 35	480 ± 170	100 ± 38	380 ± 46	330 ± 66	200 ± 57	300 ± 58	190 ± 47	434 ± 110
**56ME*SS***	10 ± 1.6	35 ± 6.4	93 ± 44	76 ± 57	21 ± 2	29 ± 1	4.6 ± 0.4	59 ± 4	66 ± 22	13 ± 2	16 ± 1	13 ± 2	36 ± 10
** *Platinum(IV) Precursors* **													
**PHEN*SS*(IV)(OH)_2_**	710 ± 300	4900 ± 610	16,000 ± 4500	800 ± 84	1700 ± 200	4300 ± 530	310 ± 92	3000 ± 530	1700 ± 350	3400 ± 2200	1700 ± 200	1300 ± 350	3318 ± 880
**56ME*SS*(IV)(OH)_2_**	36 ± 7	190 ± 23	480 ± 140	59 ± 7	190 ± 150	120 ± 22	15 ± 2.6	240 ± 22	210 ± 45	43 ± 2.5	61 ± 7	170 ± 120	151 ± 50
** *FDA-approved Chemotherapeutics* **													
Cisplatin	11,300 ± 1900	3800 ± 1100	6500 ± 800	1000 ± 100	900 ± 200	2400 ± 300	1200 ± 100	1900 ± 200	400 ± 100	7500 ± 1300	5200 ± 520	28,000 ± 1600	5842 ± 610
Oxaliplatin	900 ± 200	1800 ± 200	500 ± 100	160 ± 100	1600 ± 100	4100 ± 500	2900 ± 400	900 ± 200	3000 ± 1200	900 ± 200	nd	800 ± 100	1463 ± 320
Carboplatin	>50,000	>50,000	>50,000	9200 ± 2900	14,000 ± 1000	24,000 ± 2200	15,000 ± 1200	19,000 ± 1200	5700 ± 200	>50,000	>50,000	>50,000	32,242 ± 1450
** *Halogenated PAA Derivatives (Axial Ligands)* **													
4-CPA	>50,000	>50,000	>50,000	>50,000	>50,000	>50,000	>50,000	>50,000	>50,000	>50,000	>50,000	>50,000	>50,000
4-FPA	>50,000	>50,000	>50,000	>50,000	>50,000	>50,000	>50,000	>50,000	>50,000	>50,000	>50,000	>50,000	>50,000
4-BPA	>50,000	>50,000	>50,000	>50,000	>50,000	>50,000	>50,000	>50,000	>50,000	>50,000	>50,000	>50,000	>50,000
4-IPA	>50,000	>50,000	>50,000	>50,000	>50,000	>50,000	>50,000	>50,000	>50,000	>50,000	>50,000	>50,000	>50,000

**Table 8 molecules-27-07120-t008:** ROS production upon treatment with **1**–**8** in HT29 colon cells at 24, 48, and 72 h. The data presented for platinum(II) (**PHEN*SS*** and **56ME*SS***), platinum(IV) (**PHEN*SS*(IV)(OH)_2_**, and **56ME*SS*(IV)(OH)_2_**) precursors were obtained from a previous study [21].

Compounds	ROS Production in Different Time Intervals (RFU)		
	24 h	48 h	72 h
Control	60 ± 5	81 ± 4	97 ± 9
Cisplatin	221 ± 10	280 ± 12	318 ± 9
TBHP	514 ± 3	336 ± 2	332 ± 5
**PHEN*SS***	174 ± 2	172 ± 9	176 ± 7
**56ME*SS***	240 ± 5	218 ± 3	255 ± 4
**PHEN*SS*(IV)(OH)_2_**	144 ± 5	273 ± 4	303 ± 1
**56ME*SS*(IV)(OH)_2_**	259 ± 3	356 ± 11	438 ± 7
4-CPA	167 ± 13.2	207 ± 14.5	233 ± 10.4
**1**	272 ± 9	368 ± 19	409 ± 11
**2**	358 ± 21	508 ± 16	564 ± 10
4-FPA	197 ± 8	224 ± 11	236 ± 6
**3**	305 ± 28	409 ± 22	558 ± 13
**4**	579 ± 18	681 ± 17	738 ± 15
4-BPA	172 ± 9	198 ± 10	222 ± 13
**5**	281 ± 16	399 ± 12	478 ± 9
**6**	496 ± 14	598 ± 14	643 ± 14
4-IPA	137 ± 7	182 ± 12	206 ± 9
**7**	211 ± 9.1	281 ± 12	322 ± 17
**8**	345 ± 19.3	406 ± 14	466 ± 13

## Data Availability

All data relevant to the publication are included.

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
