# Peer review of "Bioactive Platinum(IV) Complexes Incorporating Halogenated Phenylacetates"

_molecules, 2022, doi:10.3390/molecules27207120_

Round 1
Reviewer 1 Report
This manuscript describes the synthesis, characterization and biological investigation of a new series of cytotoxic Pt(IV) complexes incorporating halogenated phenylacetic acid (PAA) derivatives. The characterization was made by HPLC, NMR, UV, CD and ESI-MS measurements. The reduction properties were investigated by 1H-NMR and monodimensional 195Pt-NMR spectroscopies and also studies were undertaken to determine solubility, stability and lipophilicity of the comoplexes. The biological studies conducted on a series of different cancerous cell lines confirm that the studied complexes may induce DNA damage through the ROS production inside the cancer cells, inducing apoptosis.
Apart from few typos (correct in some points in the abstract phenylacetatic with phenylacetic acid), the manuscript is complete and is well framed in the Introduction in the contest of the literature, the style is clear and the references adequate. The paper deserves publication.
Author Response
All authors would like to thank the reviewer for the time and effort in reviewing the manuscript. As for the typographical errors in the abstract section, we have corrected it.
Reviewer 2 Report
The presented work was performed at the junction of organic chemistry, chemistry of coordination compounds and medical chemistry. The authors synthesized new platinum complexes and proved their structure using nuclear magnetic resonance spectroscopy, mass spectrometry and electron spectroscopy. The methods of synthesis and research correspond to those used in this field.
The results obtained are presented correctly and accessible. The list of references corresponds to the general state of research.
The only doubt is the possibility of comparing the obtained six-coordinate complexes of platinum (IV) with derivatives of divalent platinum having a plane-square geometry.
In general, the work can be published when justifying the correctness of the comparison of platinum compounds in various degrees of oxidation and with different geometries of complexes.
Reviewer 3 Report
The manuscript of the article entitled "Bioactive Platinum(IV) Complexes Incorporating Halogenated Phenylacetates” by authors Angelico D. Aputen, Maria George Elias, Jayne Gilbert, Jennette A. Sakoff, Christopher P. Gordon, Kieran F. Scott and Janice R. Aldrich-Wright was very carefully elaborated at a high level. The prepared compounds and platinum complexes have been properly characterized using appropriate physicochemical methods. The authors tested the stability of the complexes in a model phosphate buffer environment. In my opinion, it will be necessary to further test the stability of the prepared complexes in human blood plasma and/or serum. These environments contain esterase enzymes which hydrolyze ester bonds relatively rapidly. The prepared complexes have character of esters and it is necessary to assess whether hydrolysis occurs before they are transported to the tumor cells. The present manuscript should be acceptable after minor revision for publishing in the Molecules.
